# Cyclosporine A-resistant CAR-T cells mediate antitumour immunity in the presence of allogeneic cells

Yixi Zhang[1,2,11], Hongyu Fang[1,2,11], Guocan Wang[1,2,11], Guangxun Yuan[1,2], Ruoyu Dong[3], Jijun Luo[1,2], Yu Lyu[4], Yajie Wang[5,6], Peng Li[7], Chun Zhou [8], Weiwei Yin[9,10], Haowen Xiao [3,12] ✉, Jie Sun [5,6,12] ✉ & Xun Zeng [1,2,12] ✉

Chimeric antigen receptor (CAR)-T therapy requires autologous T lymphocytes from cancer patients, a process that is both costly and complex. Universal CAR-T cell treatment from allogeneic sources can overcome this limitation but is impeded by graft-versus-host disease (GvHD) and host versus-graft rejection (HvGR). Here, we introduce a mutated calcineurin subunit A (CNA) and a CD19-specific CAR into the T cell receptor α constant (*TRAC*) locus to generate cells that are resistant to the widely used immunosuppressant, cyclosporine A (CsA). These immunosuppressant-resistant universal (IRU) CAR-T cells display improved effector function in vitro and anti-tumour efficacy in a leukemia xenograft mouse model in the presence of CsA, compared with CAR-T cells carrying wild-type CNA. Moreover, IRU CAR-T cells retain effector function in vitro and in vivo in the presence of both allogeneic T cells and CsA. Lastly, CsA withdrawal restores HvGR, acting as a safety switch that can eliminate IRU CAR-T cells. These findings demonstrate the efficacy of CsA-resistant CAR-T cells as a universal, 'off-the-shelf' treatment option.

Chimeric antigen receptor (CAR)-T therapy has been heralded a promising era of cancer treatment over the past decades. However, current autologous CAR-T manufacture can be costly, time-consuming[1], and may fail to meet therapeutic requirements due to the insufficient numbers of patient T cells[2]. However, these limitations could be overcome by allogeneic CAR-T therapy. Two key problems should be solved in the application of allogeneic CAR-T cell therapy: graft-versus-host diseases (GvHD)[3,4] and host-versus-graft rejection (HvGR)[5]. Great progress has been made to minimize the unwanted GvHD by ablating αβ TCR[6–8], using γδ T cells[9–11], selecting specific T cells with non-self antigens[12–14], or disrupting αβ TCR expression[8]. However, HvGR, which shortens the persistence of

[1]State Key Laboratory for Diagnosis and Treatment of Infectious Diseases, National Clinical Research Center for Infectious Diseases, National Medical Center for Infectious Diseases, Collaborative Innovation Center for Diagnosis and Treatment of Infectious Diseases, The First Affiliated Hospital, Zhejiang University School of Medicine, Hangzhou 310003, China. [2]Research Units of Infectious disease and Microecology, Chinese Academy of Medical Sciences, Hangzhou 310003, China. [3]Department of Hematology, Sir Run Run Shaw Hospital, Zhejiang University School of Medicine, Hangzhou 310016, China. [4]Zhejiang University-University of Edinburgh Institute (ZJU-UoE Institute), Zhejiang University School of Medicine, International Campus, Zhejiang University, Hangzhou 310058, China. [5]Bone Marrow Transplantation Center of the First Affiliated Hospital and Department of Cell Biology, Zhejiang University School of Medicine, 866 Yuhangtang Road, Hangzhou 310058, China. [6]Liangzhu Laboratory, Zhejiang University Medical Center, Hangzhou 311121, China. [7]Puluoting Health Technology Co., Ltd, Hangzhou 310003, China. [8]School of Public Health & Sir Run Run Shaw Hospital, Zhejiang University School of Medicine, Hangzhou 310016, China. [9]Key Laboratory for Biomedical Engineering of the Ministry of Education, College of Biomedical Engineering and Instrument Science, Zhejiang University, Hangzhou 310058, China. [10]Department of Thoracic Surgery, Sir Run Run Shaw Hospital, Zhejiang University School of Medicine, Hangzhou 310016, China. [11]These authors contributed equally: Yixi Zhang, Hongyu Fang, Guocan Wang. [12]These authors jointly supervised this work: Haowen Xiao, Jie Sun, Xun Zeng. ✉e-mail: haowenxiaoxiao@zju.edu.cn; sunj4@zju.edu.cn; xunzeng@zju.edu.cn

universal CAR-T cells and hampers the anti-tumour activities, remains a challenge for universal CAR-T therapy.

Cyclosporine A (CsA) is a first-line immunosuppressive drug that binds to cyclophilin A and inhibits the nuclear factor of activated T cell (NFAT) dephosphorylation and downstream signaling in T cells. CsA widely used in organ transplantation with limited concerns of safety. Calcineurin subunit A (CNA) forms a complex with CyPA, and mutations of certain amino acids on CNA can disrupt the docking of CsA-CyPA, which restores the NFAT dephosphorylation and subsequent T cell activation[15].

Here, we describe the developments of immunosuppressant-resistant universal (IRU) CAR-T cells, in which CAR and mutated CNA (mCNA) (V314R and Y341F) genes are inserted into the T cell receptor α constant (*TRAC*) locus to disrupt TCR expression and prevent GvHD. We show that IRU CAR-T cells retain anti-tumour effect in a CsA-induced immunosuppressant environment that HvGR is largely impaired. Furthermore, IRU CAR-T cells can be eliminated by host T cells after withdrawal of CsA if necessary, which increases the safety profile of this CAR-T therapy.

## Results

### IRU CAR-T cells retained effector functions in the presence of CsA in vitro

To confirm the resistance of the mCNA against CsA, we transduced wild type (WT)/mCNA genes into Jurkat cells with NFAT-GFP reporter gene (Jurkat-NFAT-GFP) and analysed GFP expression that represented the activation of NFAT signaling pathway[16], upon Nalm6 tumour cell line expressing luciferase and GFP (Nalm6-FFLuc-GFP) stimulation. Jurkat-NFAT-GFP cells with mCNA expressed higher frequency of GFP+ cells than WT control in a CsA dose-dependent manner, revealing the resistance of mCNA to CsA-inhibited NFAT signaling (Supplementary Fig. 1a). Next, to construct WT/IRU CAR in primary T cells, we directed CAR cassette (P2A-1928z CAR-T2A-WT/mCNA) using AAV6 into the *TRAC* locus that was disrupted by CRISPR/Cas9 (Fig. 1a). The integration of CAR into *TRAC* locus design has been firstly proposed by Eyquem et al.[17], which contributed a better anti-tumour effect of CAR-T cells and was followed by several studies[18]. The high selectivity for *TRAC* locus integration and the low potential off-target hotspots have been validated previously, confirming the precision of CRISPR/Cas9

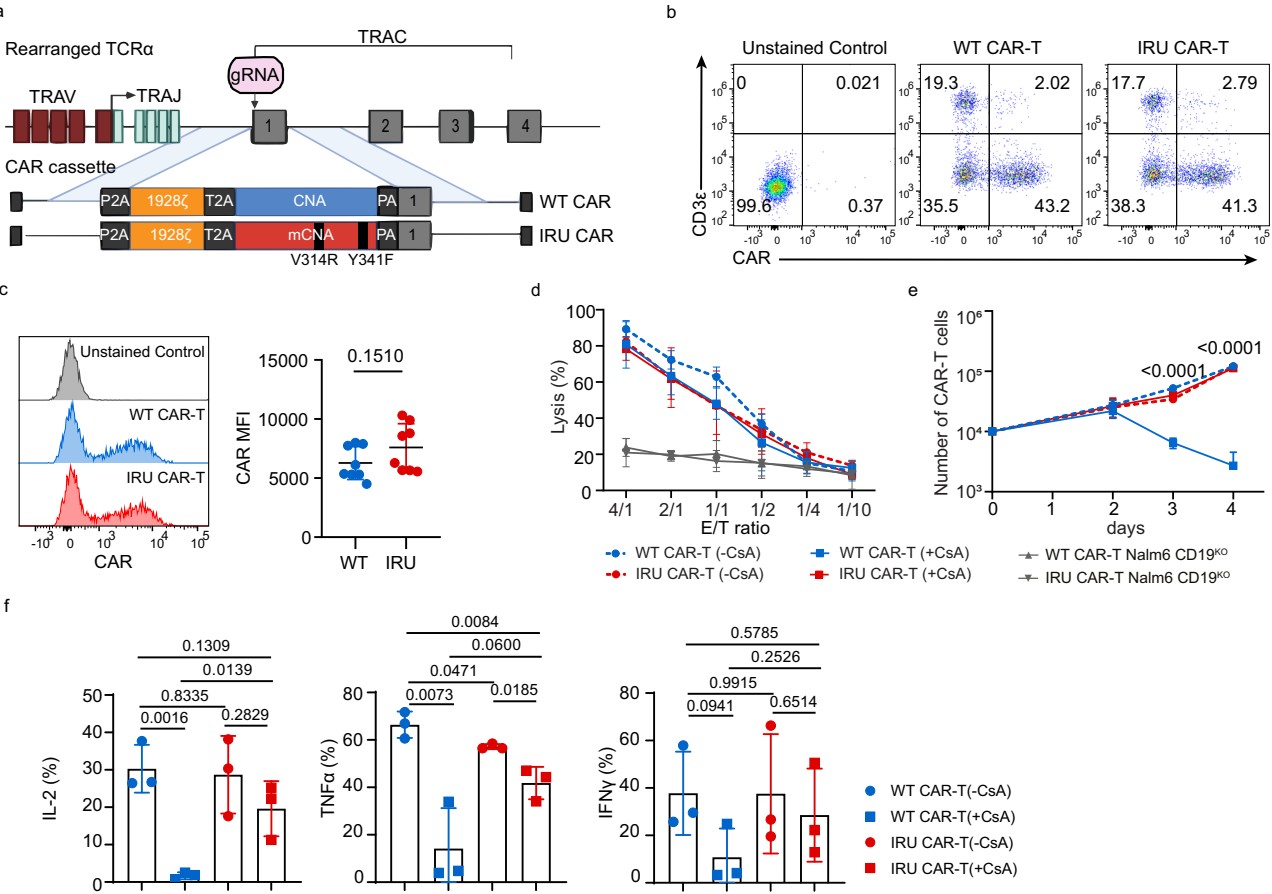

**Fig. 1 | IRU CAR-T cells retained effector functions in the presence of CsA in vitro. a** schematic of CRISPR/Cas9-targeted WT/IRU CAR gene incorporation into the 1st exon of *TRAC* locus. Top, *TRAC* locus; bottom, rAAV6 containing the WT/IRU CAR repertoire flanked by homology arms of inserted region. The CNA of IRU CAR was mutated at two points (V314R, Y341F) relative to the WT CAR.
**b**, **c** Representative flow plots (**b**), the histogram analysis (**c**, left), and mean fluorescence intensity (MFI) (**c**, right, *n* = 8 biologically independent samples) of WT/IRU CAR expression on human primary T cells after 4 days of gene targeting.
**d** Cytotoxic activity using Nalm6-FFLuc-GFP as target cells (*n* = 3 biologically independent samples) in 24 hrs cytotoxicity assay at different E:T ratios. Target cell lysis rate was calculated by (1- (RLUsample)/(RLUmax)) ×100 (RLU: relative luminescence units). CD19^KO Nalm6-FFLuc-GFP was used as a negative control to validate the specificity of CAR (*n* = 3 biologically independent samples). **e** Proliferation

of CAR-T cells upon the stimulation with 10-fold irradiated (50 Gy) Nalm6-FFLuc-GFP (*n* = 3 biologically independent samples). **f** Cytokines (IL-2, IFNγ, and TNFα) expression of CAR-T cells upon stimulation with 10-fold irradiated (50 Gy) Nalm6-FFluc-GFP (*n* = 3 biologically independent samples). All data are mean ± s.d. *P* values were determined by two-tailed Unpaired t test (**c**, **f**) or Multiple t tests adjusted by the Holm-Sidak method (**e**). WT CAR-T (-CsA) as blue circle, WT CAR-T (+CsA) as blue square, IRU CAR-T (-CsA) as red circle and IRU CAR-T (+CsA) as red square. *TRAV* T cell receptor alpha variable, TRAJ T cell receptor joining, *TRAC* T cell receptor alpha constant, (m)CNA (mutated) calcineurin subunit A, gRNA guide RNA, 2A-self-cleaving peptide-based multi-gene expression, P2A porcine teschovirus1 2A, T2A *Thosea asigna* virus 2A, V314R valine 314 arginine, Y341F tyrosine 341 phenylalanine.

mediated genome editing method used[17,19]. A high knock-in frequency was achieved with a high multiplicity of infection (MOI) ratio for AAV6 (Fig. 1b and Supplementary Fig. 1b). The proliferation of TCR⁻ CAR-T cells can be further supported in the presence of cytokines[18,20,21], which finally achieved the required cell quantity. Furthermore, we characterised the mean fluorescence intensity (MFI), the ratio of CD4/CD8 and differentiation status (expression of CD45RA and CCR7) in IRU and WT CAR-T cells from different donors. No significant phenotype difference was observed between IRU and WT CAR-T cells (Fig. 1c and Supplementary Fig. 1c, d) at this stage.

We performed functional assays of IRU CAR-T cells in terms of cytotoxicity, proliferation, and cytokine secretion in vitro with 300 ng/ml CsA, which is the effective blood concentration of CsA in clinical treatment[22,23]. For cytotoxicity assay, CAR-T cells were stimulated with Nalm6-FFLuc-GFP at different ratios of effector and target cells (E:T) for 24 hrs. Interestingly, there was no significant difference for killing Nalm6-FFluc-GFP cells between WT CAR-T (+CsA) and IRU CAR-T (+CsA) groups. In addition, WT/IRU CAR-T could not kill Nalm6-CD19^KO with the increasing of E/T ratio. This result showed the incapability of CAR-T cells when the target is missing, indicating the specificity of CAR-T cells against tumour target (Fig. 1d). However, in proliferation assay, IRU CAR-T (+CsA) group showed a comparable trend as IRU/WT CAR-T (-CsA) groups upon the stimulation with irradiated Nalm6-FFLuc-GFP, whereas the number of WT CAR-T (+CsA) cells gradually reduced within 4 days (Fig. 1e). In addition, the intracellular cytokine staining showed that the percentages of IL-2-, IFNγ-, TNFα-secreting CAR-T cells lost or dramatically decreased in WT CAR-T group (+CsA) after being stimulated by irradiated Nalm6-FFLuc-GFP for 24 hrs. In contrast, IRU CAR-T (+CsA) cells significantly maintained to secret IL-2, IFNγ, and TNFα (Fig. 1f and Supplementary Fig. 1e), suggesting the mCNA was resistant, although not completely, to CsA inhibitory effect. In addition, we tested whether the alteration of phenotype and cytokine was dependent on either CAR or CsA. We generated a GPC3 specific CAR-T (GPC3-CAR-T) and a CD19-specific CAR-T (CD19-CAR-T, i.e. WT/IRU CAR-T in this paper) with WT CNA or mCNA from the same donor and compared the difference of phenotype and cytokine production between them. These CAR-T cells were stimulated with either HepG2-FFluc-GFP (GPC3⁺) or Nalm6-FFLuc-GFP cells (CD19⁺) for 24hrs, respectively, and performed FACS to check phenotypes (Supplementary Fig. 2a, b). Different CAR-T cells displayed different phenotypes with/without CsA. In the perspective of cytokine, the trend of cytokine production of GPC3-CAR-T cells was the same as CD19-CAR-T cells with/without CsA (Supplementary Fig. 2c), i.e., the addition of CsA would significantly abolish the cytokine production of WT GPC3 CAR-T cells, but not IRU GPC3 CAR-T cells. However, the detailed frequency of cytokine-secreting CAR-T cells were not the same (Supplementary Fig. 2b) in the absence of CsA for different CAR-T cells. These data suggested that different CAR-T cells may affect phenotype. However, CsA resistant mechanism drove a similar pattern of cytokine production.

Further, we investigated the transcriptome of CD8⁺ IRU and WT CAR-T cells with/without CsA upon the stimulation of Nalm6-FFLuc-GFP for 24 hrs. The principal component analysis (PCA) depicted the transcriptomic profile of IRU CAR-T (+CsA) group was similar to those of other groups without CsA, but unique to that of WT CAR-T (+CsA) (Fig. 2a). Furthermore, we used Kyoto encyclopedia of genes and genomes (KEGG) pathway methodology to analyse the differentially expressed genes (DEGs). Compared to those of WT CAR-T (+CsA), upregulated DEGs in IRU CAR-T (+CsA), as well as in WT/IRU CAR-T (-CsA), were enriched in cell proliferation pathways including cell cycle and DNA replication, while downregulated DEGs were concentrated in cell apoptosis and allograft rejection pathways (Fig. 2c and Supplementary Fig. 3a). In addition, upregulated DEGs related to cell cycle (e.g., PCNA[24,25], CDK4[26,27] and CDC25A[28]) and effector function (e.g., IFNG[29], CSF2, Gzmb[29], TNF[29], IL2[29], IL3[29] and IL5[29]), and downregulated

DEGs related to cell apoptosis (e.g., NTRK1[30] and ATM[31]) were observed in IRU CAR-T cells (+CsA) compared to those in WT CAR-T cells (+CsA) (Fig. 2b, d and Supplementary Fig. 3b, c). Series of qPCR assay were performed to validate the reliability of RNA-seq, which showed the consistent trend of gene expression in the four groups (Fig. 2e). Taken together, these results indicate that IRU CAR-T cells expand and maintain their cytokine secretion and cytotoxicity in the CsA-induced immunosuppressive environment in vitro.

However, it was worthy to note that the expressions of many genes were lower in IRU CAR-T compared to that in WT CAR-T in the absence of CsA, which raised a very important question that whether mCNA affects T cell functions without CsA treatment. To address this question, we firstly compared the GFP expression that represented the activation of NFAT signaling pathway[16] in WT/IRU Jurkat-NFAT-GFP cell line, upon the simulation of Nalm6 cell line without CsA. It was observed that GFP⁺ cells in WT CAR-T-Jurkat-NFAT-GFP cell line were higher than that in IRU-CAR-T-Jurkat-NFAT-GFP cell line in the absence of CsA (Supplementary Fig. 4a). We hypothesised that this phenomenon was due to the less efficient dephosphorylation of NFAT in mCNA compared to WT CNA. To confirm this, phosphor-flow was performed to detect the level of NFAT phosphorylation in primary T cells. Similarly, we found that the MFI of the phosphorylated NFATc1 in WT CAR-T cells was lower than that in IRU CAR-T cells without CsA, but the MFI of the phosphorylated NFATc1 in WT CAR-T cells was higher than in IRU CAR-T cells with CsA (Supplementary Fig. 4b, c). Collectively, these data indicated that mCNA had less efficient for NFAT dephosphorylation which affected downstream genes of NFAT signaling pathway and T cell effector functions.

## IRU CAR-T cells would not induce GvHD in vitro and in vivo

As the IRU CAR-T was designed as a universal CAR-T, whether it would induce GvHD should be re-confirmed in this system, although the deficiency of TCR had been proved to significantly diminish GvHD in clinical trial[8]. IRU CAR-T cells were cocultured with irradiated recipient PBMCs for 4 days in vitro. Comparing CAR-T cells that with functional TRAC locus (conventional CAR-T, made by lentivirus infection), IRU CAR-T cells had significantly lower frequency of IFNγ producing cells (Supplementary Fig. 5a-b). This result indicated the knockout of TCR in IRU CAR-T cells significantly inhibited alloreactive CAR-T cell responses against recipient PBMC cells in vitro. However, the addition of CsA abolished GvHD as the activation of T cells was inhibited (Supplementary Fig. 5a, b). In in vivo assay, conventional CAR-T and IRU CAR-T cells were injected in NOD-Prkdc^scid Il2rg^em1/Smoc mice (NSG) mice, respectively. The weight and GvHD score (assessed by parameters including posture, activity, fur, skin and weight loss as previously described[32]) of mice were recorded. The in vivo experiments excluded the circumstances with CsA, as CsA might induce GvHD-like symptoms such as weight change to interfere the outcome interpretation. Compared to IRU CAR-T group, mice injected with conventional CAR-T cells significantly lost weight and scored higher in GvHD index after 5 weeks (Supplementary Fig. 5c, d). The in vivo result consistently supported that the GvHD would significantly decrease due to the lack of TCR expression in IRU CAR-T cells.

## IRU CAR-T cells maintained anti-tumour activity in the presence of CsA in a xenograft leukemia mouse model

Further, the anti-tumour efficacy of IRU CAR-T cells in the CsA circumstances was validated in vivo. In an acute ALL NSG mouse model, CAR-T cells were transferred 4 days after Nalm6-FFluc-GFP inoculation, and CsA was injected daily from day 3 to day 21 and every other day thereafter (Fig. 3a). Both WT and IRU CAR-T cells could successfully eliminate tumour cells and contribute to better survival of mice without CsA (Fig. 3b, c and Supplementary Fig. 6). However, compared to the WT CAR-T (+CsA) group, IRU CAR-T (+CsA) group could control leukemia progression with better survival (Fig. 3b, c and

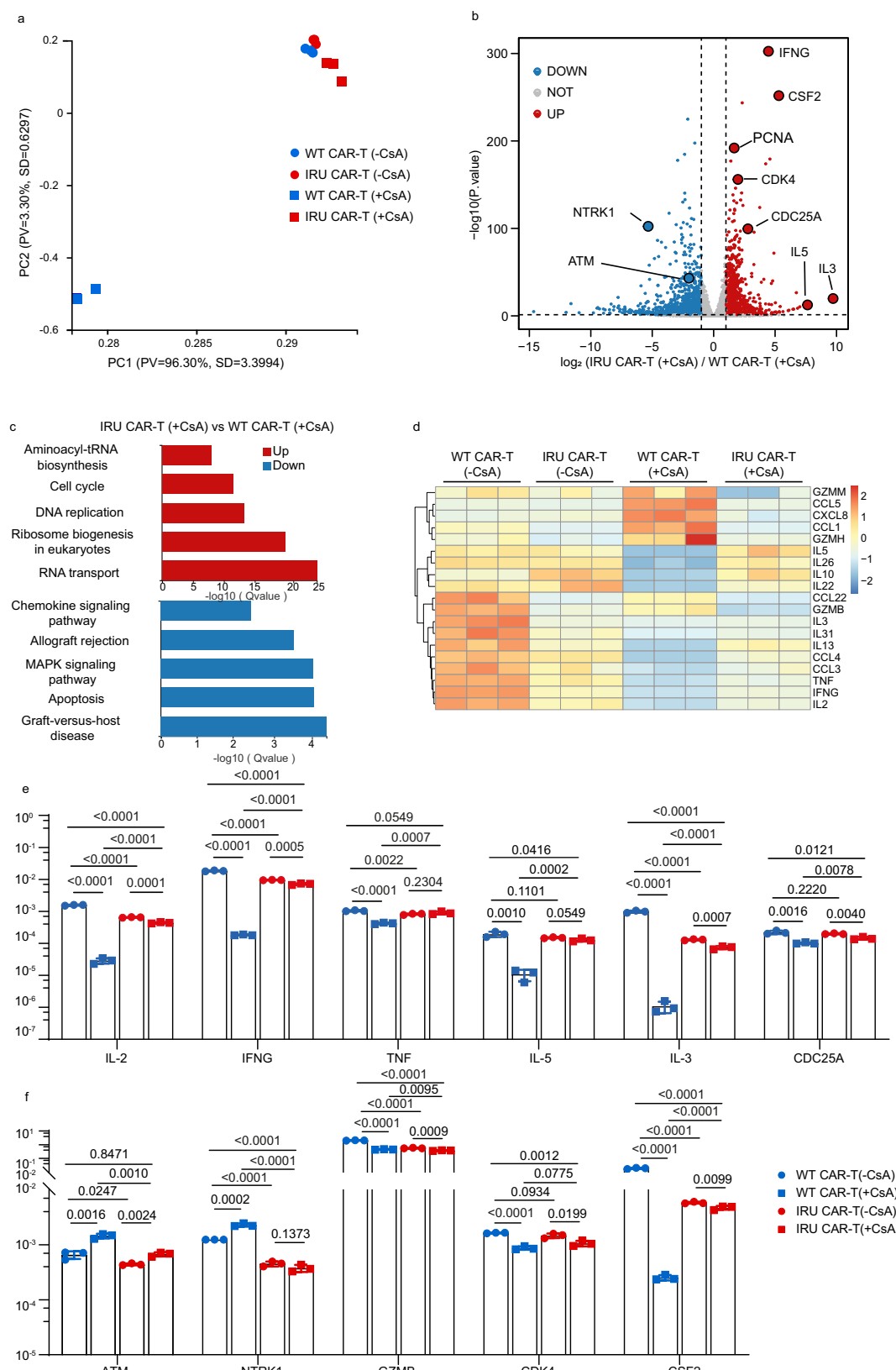

Supplementary Fig. 6). To further evaluate the CAR-T cell presence and function in vivo, we established ex vivo experiments to investigate the kinetics of CAR-T cell at different time points. At day 14 with CsA, the percentage of IRU CAR-T was greater than that of WT CAR-T showed a significant difference, suggesting improved survival or proliferation (Fig. 3d and Supplementary Fig. 7a and b). Notably, cytokine production in the detectable WT CAR-T (+CsA) cells was significantly impaired, as evidenced by low frequencies of cytokine-secreting cells compared with the IRU CAR-T (+CsA) group at day 14 (Fig. 3e, f and Supplementary Fig. 7d, e). In addition, the WT CAR-T (+CsA) group showed fewer central memory T cells ($T_{CM}$) compared with the other groups at day 14 (Supplementary Fig. 7c). These data suggest that IRU

**Fig. 2 | The transcriptional signatures IRU CAR-T cells in the presence of CsA in vitro. a–d** Bulk RNA Sequencing analysis for four groups (WT/ IRU CAR-T cells (+/- CsA), n=3 technical replicates, 1 donor upon stimulation with 10-fold irradiated (50 Gy) Nalm6-FFluc-GFP. **a** PCA depicted the clustering of the transcriptome profile of WT/IRU CAR-T cells (+/- CsA). **b** Volcano plots showed the DEGs in WT/IRU CAR-T (+CsA) groups. **c** KEGG analysis showed the up- and downregulated gene sets in WT/IRU CAR-T (+CsA) groups. **d** Heatmap demonstrated the DEGs in cytokine and chemokine among the WT/IRU CAR-T cells with/without CsA. **e, f** qPCR validation of representative genes in bulk RNA-seq (*n* = 3 technical replicates, 1 donor. Data from the other two donors can be found in the source data file). All data are means ± s.d. *P* values were determined by two-tailed Unpaired t test (**e, f**). WT CAR-T (-CsA) as blue circle, WT CAR-T (+CsA) as blue square, IRU CAR-T (-CsA) as red circle and IRU CAR-T (+CsA) as red square.

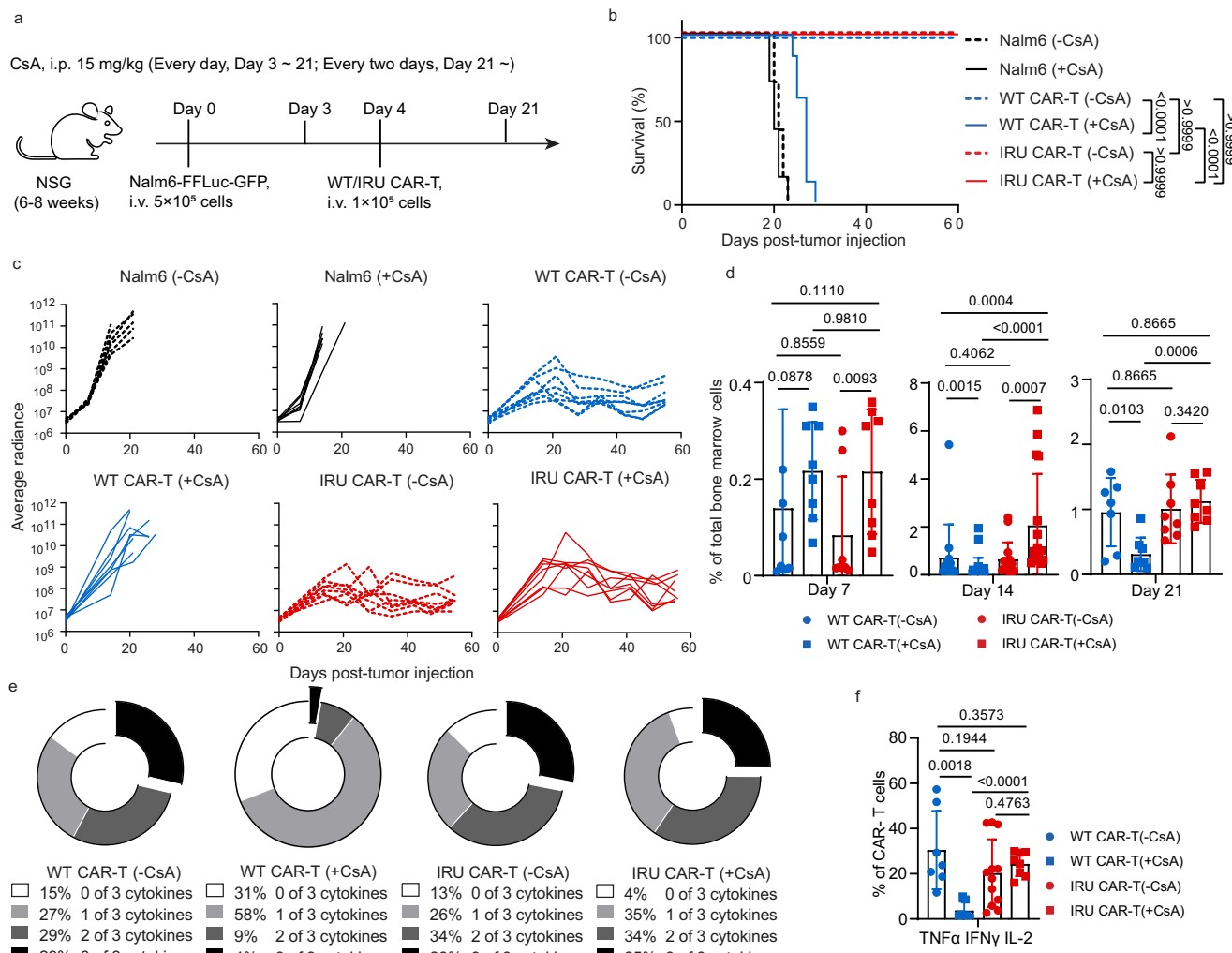

**Fig. 3 | IRU CAR-T cells retained anti-tumour functions in the presence of CsA in vivo. a** Schematic of NOD.Cg-*Prkdc^scid^Il2rg^em1Smoc^* (NSG) (female, aged 8-12 weeks) model for (**b**−**e**). **b** Kaplan−Meier analysis of the mice survival in each experimental group (*n* = 8 animals in 2 batches). **c** Tumour burden (average radiance, ph/s) of mice treated with WT/IRU CAR-T cells (+/- CsA) (*n* = 8 animals in 2 batches).
**d** Frequencies of CAR-T cells in bone marrow after tumour injection at day 7, 14 and 21 (*n* = 8 animals in two batches, 14 animals in three batches, and 7 animals in two batches at day 7, 14, and 21 respectively in WT CAR-T (-CsA); n = 8 animals in two batches, 24 animals in four batches and 8 animals in two batches at day 7, 14 and 21 respectively in WT CAR-T (+CsA); n = 8 animals in two batches, 15 animals in three batches and 8 animals in two batches at day 7, 14 and 21 respectively in IRU CAR-T (-CsA); n = 8 animals in two batches, 17 animals in four batches and 8 animals in two batches at day 7, 14 and 21 respectively in IRU CAR-T (+CsA)). **e, f** Cytokine detection of CAR-T cells in each group at day 14 upon stimulation with PMA/Ionomycin for 4 hrs. Donut (**e**) and column charts (**f**) indicated the proportion of CAR-T cells in each group that expressed cytokines (*n* = 8 animals in two batches in each group). All data are means±s.d. *P* values were determined by Mantel-Cox log-rank test (**b**) or Mann-Whitney test (**d**) or Multiple t test adjusted by the Holm-Sidak method (**f**). WT CAR-T (-CsA) as blue circle, WT CAR-T (+CsA) as blue square, IRU CAR-T (-CsA) as red circle, and IRU CAR-T (+CsA) as red square. Abbreviations: i.v. intravenous injection; i.p. intraperitoneal injection.

CAR-T cells could retain their abilities to diminish leukemia cells in the presence of CsA in vivo.

### IRU CAR-T cells can avoid allogeneic T cell-induced rejection in vitro with CsA

Due to the central role of allogeneic recipient T cells (RTCs) in induction of HvGR[33], we established mixed lymphocyte reaction (MLR) models to mimic RTC rejection[34], using HLA-A2 as a marker to distinguish two donors. HLA-A2⁻ donor CAR-T cells were co-cultured with a 4-fold excess of HLA-A2⁺ RTCs with the stimulation of irradiated Nalm-6-FFluc-GFP-β2m^KO^ cells that depleted MHC-I to avoid nonspecific allogeneic rejection (Fig. 4a).

CAR-T cells in all groups, except in WT CAR-T cells (+CsA), could proliferate without RTCs (Supplementary Fig. 8a). However,

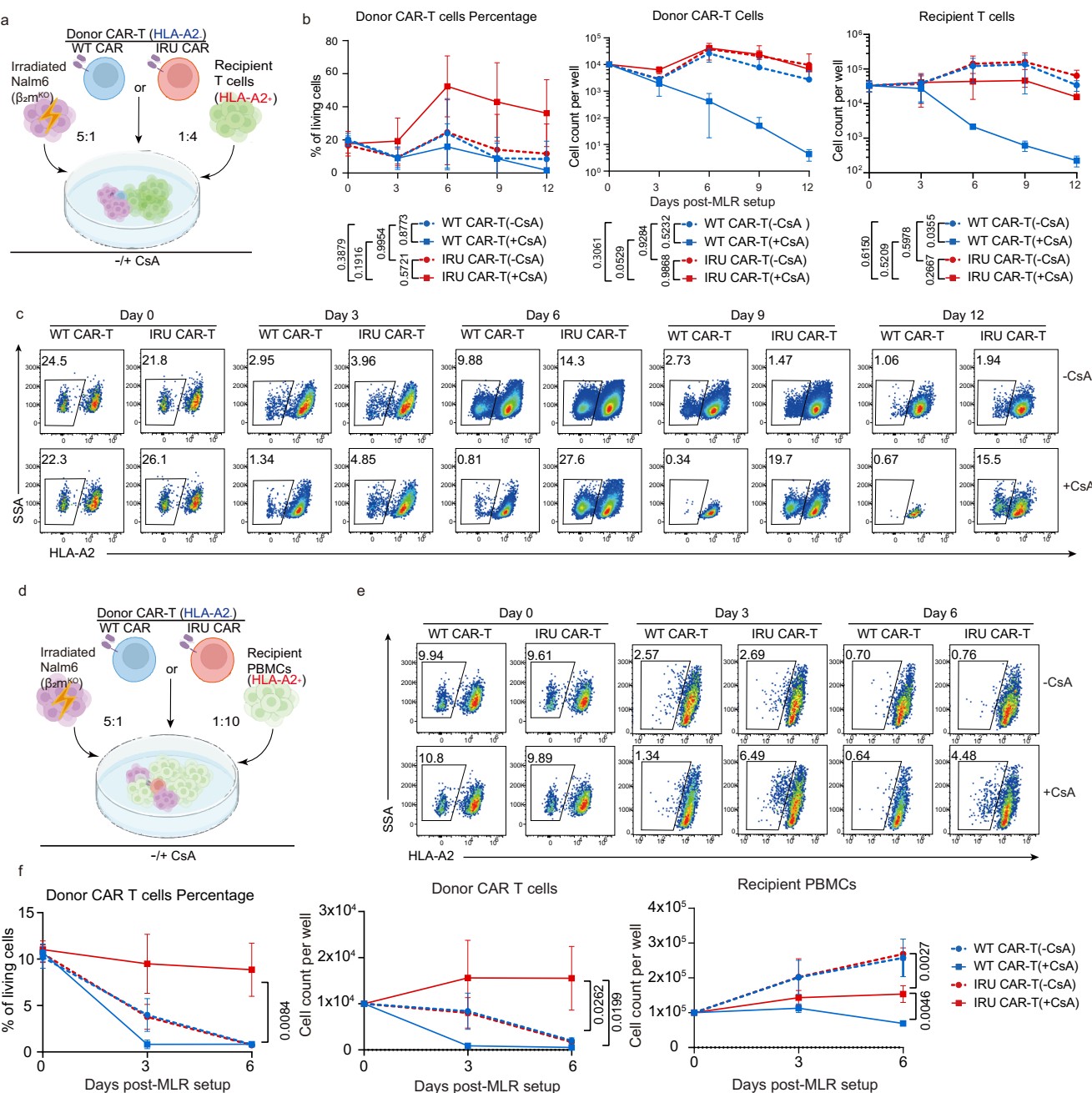

**Fig. 4 | The survival of IRU CAR-T cells is not reduced in the presence of recipient T cells with CsA in vitro. a** Schematic of MLR-RTCs for **b**, **c** ($n$ = 3 biologically independent samples). Donor CAR-T (WT/IRU CAR, HLA-A2⁻) were mixed with 4-fold recipient T cells (HLA-A2⁺) and 5-fold irradiated (50 Gy) Nalm6-FFluc-GFP-β₂mᴷᴼ with or without CsA. **b**, **c** Curve chart showing percentage of donor CAR-T cells at day 0, 3, 6, 9, and 12 (**b**, left), absolute cell counts of donor CAR-T cells (**b**, middle) and recipient T cells (**b**, right), and representative flow plots (**c** top: -CsA group; bottom: +CsA group). *P* values means the significance among four groups at day 12. **d** Schematic MLR- PBMCs for **e**, **f** ($n$ = 3 biologically independent samples).

Donor CAR-T (WT/IRU CAR; HLA-A2⁻) were mixed with 10-fold recipient PBMCs (HLA-A2⁺) and 5-fold irradiated (50 Gy) Nalm6-FFluc-GFP-β₂mᴷᴼ with or without CsA. **e**, **f** Representative flow plots (**e**, top: -CsA group; bottom: +CsA group) and a curve chart (**f**, left) showing percentage of donor CAR-T cells at day 0, 3 and 6, and curve charts of absolute cell counts of donor CAR-T cells (**f**, middle) and recipient PBMCs (**f**, right). *P* values means the significance among four groups at day 6. All data are means±s.d. *P* values were determined by Multiple t test adjusted by the Holm-Sidak method (**c** and **f**). WT CAR-T (-CsA) as blue circle, WT CAR-T (+CsA) as blue square, IRU CAR-T (-CsA) as red circle and IRU CAR-T (+CsA) as red square.

the addition of RTCs resulted in the different change of percentage and cell number of CAR-T cells in different groups. Prior to co-culturing with donor CAR-T cells, RTCs were primed as described in methods. Primed RTCs displayed higher percentage of IFNγ producing RTCs (Supplementary Fig. 8b), which contributed to the allo-rejection phenomenon in MLR assay. With the addition of RTCs, IRU CAR-T (+CsA) group showed a significantly higher CAR-T percentage than the rest of three groups (Fig. 4b, c). Compared to WT CAR-

T (+CsA) group, the higher percentage of CAR-T cells in IRU CAR-T (+CsA) was largely due to the incapability of WT CAR-T proliferation with CsA (Supplementary Fig. 8a). Compared to WT/IRU CAR-T cells in the absence of CsA, the higher percentage of CAR-T cells in IRU CAR-T (+CsA) might result from 1) the allo-rejection of RTCs against CAR-T cells, which resulted in the decreasing cell number of CAR-T cells; 2) the increasing number of RTCs further diluted the percentage of CAR-T cells.

Notably, not all RTCs were real allo-reactive T cells. The percentage of real allo-reactive T cells, i.e. the cells that could produce IFNγ after priming (Supplementary Fig. 8b), was not high. Meanwhile, potential interleukin produced by the interaction between CAR-T cells and Nalm6 cells/alloreactive T cells could promote the proliferation of non-alloreactive RTCs. Therefore, the decrease of CAR-T cells was not correlated linearly with the increase of RTCs.

To further validate the abilities of IRU CAR-T cells in a more clinically-relevant condition, we performed an extended MLR assay between HLA-A2⁻ donor CAR-T cells and equal quantity of HLA-A2⁺ recipient peripheral blood mononuclear cells (PBMCs) (Fig. 4d). Similar with the aforementioned MLR assay, only IRU CAR-T (+CsA) group had the highest donor CAR-T percentage and absolute cell number (Fig. 4e, f). Meanwhile, recipient PBMCs showed better survival in the absence of CsA. These data also supported that IRU CAR-T could survive better than any other group in the presence of allo-PBMCs by using CsA. Consistently, the same trend was also observed in different pairs and different ratio of donor CAR-T and recipient PBMCs (Supplementary Fig. 9a–d). Taken together, these data suggested that IRU CAR-T maintained better survival in the context of existing allo-T/PBMCs with CsA.

### IRU CAR-T cells retained anti-leukemia functions in the presence of RTCs in vivo by using CsA

Next, we established an aforementioned xenograft leukemia mouse model but using Nalm-6-FFluc-GFP-β2m$^{KO}$ cells, with the additional injection of RTCs at day 2 (Fig. 5a). Consistent with the in vitro results, only IRU CAR-T (+CsA) group could control leukemia progression and increased the survival rate of mice (Fig. 5b, c and Supplementary Fig. 10), while WT CAR-T (+CsA) could not. Of note, the WT/IRU CAR-T (-CsA) group controlled tumours poorly in this ALL model with RTCs, in contrast with effective elimination of tumour in the normal ALL model without RTCs (Fig. 3b,c). These results suggested that WT/IRU CAR-T cells might be rejected by RTCs. On the contrary, IRU CAR-T can better control tumour progression in ALL model with RTCs, suggesting the potential RTC resistance of IRU CAR-T with the protection of CsA in vivo. Next, we extracted mouse bone marrow cells at day 17, 24, and 31 to detect the presence and functions of CAR-T cells. At all specified time points, IRU CAR-T (+CsA) showed significantly higher CAR-T percentage than that in any other group, but lower RTC percentage than that in WT/IRU CAR-T (-CsA) group (Fig. 5d, e, Supplementary Fig. 11a). This result suggested that IRU CAR-T cells had better survival in the CsA induced immunosuppressant environment, as CsA might inhibit the RTC induced allorejection. In contrast, although RTCs in WT CAR-T (+CsA) group remained low level, the donor CAR-T cells in WT CAR-T (+CsA) group, regardless of CsA or RTC-induced inhibition, displayed at a significantly lower level in such condition. To further evaluate CAR-T cell function in the presence of RTCs ex vivo, we examined the kinetics of the cytokine secretion of these CAR-T cells. Consistent with the previous results, IRU CAR-T (+CsA) group showed a significantly higher frequency of IFNγ⁺TNFα⁺-secreting CAR-T cells than that in WT CAR-T (+CsA) group (Fig. 5f, g and Supplementary Fig. 11c), supporting that IRU CAR-T could better control tumour progression with CsA. Compared to the WT/IRU CAR-T (-CsA) group, the IRU CAR-T (+CsA) group exhibited higher frequency of CAR-T$_{CM}$ and lower frequency of CAR-T$_{EMRA}$ cells (except on day 24 compared with IRU CAR-T (-CsA)) at different time points (Supplementary Fig. 11b). Taken together, these results revealed that IRU CAR-T had anti-leukemia effect and avoid RTC induced rejection in the presence of CsA.

As Cas9/gRNA gene editing in cells has the potential risk for tumourigenesis[35], a safety switch in genetically manipulated CAR-T cells in vivo would be beneficial. We hypothesised that CsA withdrawal could allow the recipient to restore its immune function and eliminate IRU CAR-T cells, thus acting as a safety switch. Since CAR-T cells were hardly detected at late time points after infusion, we set up a mouse model as above with additional rechallenged RTCs at 7 days after tumour transfer and divided mice into two groups: CsA continuous injection (+CsA+CsA) group and CsA removal (+CsA-CsA) group (Fig. 6a). We investigated the frequencies of CAR-T cells and RTCs at day 14 and 21, and revealed that the frequency of CAR-T cells was significantly decreased and RTCs were significantly increased in the '+CsA-CsA' group (Fig. 6b and Supplementary Fig. 12a, b), suggesting that the CsA removal recovered the RTC triggered CAR-T cell rejection.

## Discussion

In the present study, we show that IRU CAR-T cells could retain proliferation, cytotoxicity, cytokine secretion and anti-tumour activity in the presence of CsA both in vitro and in vivo. In addition, CsA could prevent the HvGR against IRU CAR-T cells Combined with the knockout of αβ TCR, IRU CAR-T therapy can mitigate both GvHD and HvGR, which offered a promising approach to generating universal CAR-T cells.

Current strategies to solve HvGR focused on making either CAR-T cells or the host immune system "invisible" to each other. As HvGR mainly resulted from HLA mismatch[36], the disruption of HLA could prevent CAR-T cells from being recognized by host CD8⁺ T cells[37–39]. However, such strategy potentially caused two issues: firstly, it might give rise to NK-mediated rejection, as HLA-I originally interacted with NK inhibitory receptors and its deficiency would contribute to the activation of NK cells; secondly, the mere absence of HLA-I might not prevent CD4⁺ T cell-dependent rejection, as CD4⁺ T cell recognized HLA-II and orchestrated host rejection by direct killing or recruiting other immune cells[40]. An alternative approach to eliminate HvGR was to suppress the host immune system. Lymphodepletion is applied in the clinic to reduce the risk of HvGR. Similarly, the use of alemtuzumab (anti-CD52) to deplete B and T cells and protect allogeneic CD52$^{KO}$ CD19 CAR-T cells from host rejection is currently being evaluated[6–8]. These approaches used agents to deplete patient immune cells to avoid allo-rejection during the remedy. However, IRU CAR-T therapy would have a number of advantages over such approaches. Firstly, the time span of depleted immune system might not be long enough. For example, lymphodepletion was transient, which mean that HvGR would happen in longer-term especially after host immune system recovered[41,42]. In contrast, IRU CAR-T therapy could effectively resist HvGR during the whole course of treatment. Secondly, the toxicities of both lymphodepletion and CD52 were quite high. They significantly hampered the host immune system by depleting major immune cells, however, CsA only suppressed the activation of T cells. For example, T$_{CM}$ and some populations of B cells (e.g., CD5⁺ and naïve B cells) could not recover even after 2 decades by using alemtuzumab[43]. Recently, to avoid these limitations, universal CAR-T cell expressing 4-1BB ligands to deplete the alloreactive T cells instead of total T cells have been developed[34]. However, such CAR-T cells might attack host 4-1BB⁺ immune and non-immune cells, such as fibroblasts and endothelial cells[44], to induce GvHD.

Furthermore, IRU CAR-T has an additional advantage that it could be cleared after clinical remission, which added safety profile like suicide gene system[45,46]. This would avoid the potential risk of tumourigenesis that resulted from gene editing[35]. In addition, most CAR-T cells for hematological cancer therapy were targeted on lineage-specific rather than tumour-specific antigens. For example, CAR-T cells targeting CD19 would eliminate both tumour and normal B cells, and induce B cell aplasia for a long time in acute leukemia[47]. This problem was more serious in T or myeloid-derived tumour therapy, as depleted normal T cells or myeloid cells are essential for host immune defense. Therefore, this safety switch is critical in these conditions, as CAR-T cell clearance after immunotherapy would allow the regeneration of normal, functional immune system.

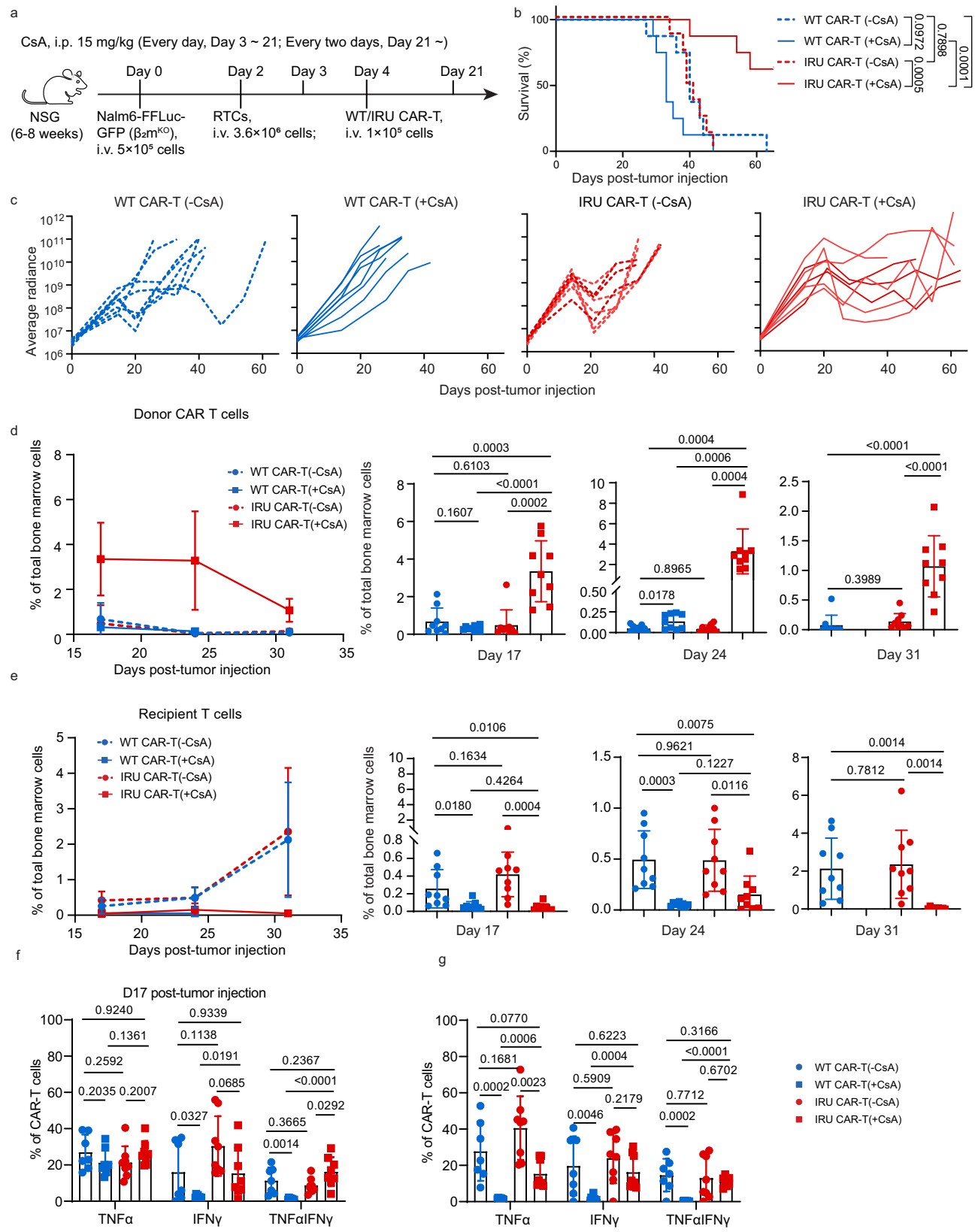

Nevertheless, there are several limitations to acknowledge regarding the potential use of IRU CAR-T cells. The use of CsA as an immunosuppressant may weaken host immunity and result in a higher risk of infections and viral reactivation events[48]. This could be overcome by limiting the duration of CsA administration to allow recovery of host immunity following tumour clearance. Another limitation of this study was that current mCNA were not fully resistant to CsA inhibition; this may require further engineering approaches to find better CNA/CNB mutations for IRU CAR-T therapy. Despite these considerations, IRU CAR-T cells may still represent a potential therapeutic avenue, which should be further evaluated for safety and efficacy.

**Fig. 5 | IRU CAR-T cells retained anti-leukemia functions in the presence of recipient T cells in vivo by using CsA. a** Schematic of ALL NOD.Cg-*Prkdc^scid^Il2rg^em1Smoc^* (NSG) (female, aged 8-12 weeks) model with RTCs for (**b**–**g**). **b** Kaplan–Meier analysis of the mice survival in mice treated with WT CAR-T cells (+/-CsA) and IRU CAR-T cells (+/-CsA) (*n* = 8 animals in two batches). **c** Tumour burden (average radiance, ph/s) in each experimental group (*n* = 8 animals in two batches). **d**, **e** Frequencies of donor CAR-T cells (**d**) and recipient T cells (**e**) in bone marrow (left, curve chart; right, column charts) over time (*n* = 9 animals in two

batches). **f**, **g** Cytokine detection of donor CAR-T cells in each group at Day 17 (**f**) and 24 (**g**) upon stimulation with PMA/Ionomycin for 4 hrs (*n* = 8 animals in two batches in each group, except *n* = 7 animals in two batches in WT CAR-T (-CsA) group). All data are means±s.d. *P* values were determined by Mantel-Cox log-rank test (**b**) or two-tailed Unpaired t test (**d**, **e**) or Multiple t test adjusted by the Holm-Sidak method (**f**, **g**). WT CAR-T (-CsA) as blue circle, WT CAR-T (+CsA) as blue square, IRU CAR-T (-CsA) as red circle and IRU CAR-T (+CsA) as red square. Abbreviations: i.v. intravenous injection; i.p. intraperitoneal injection.

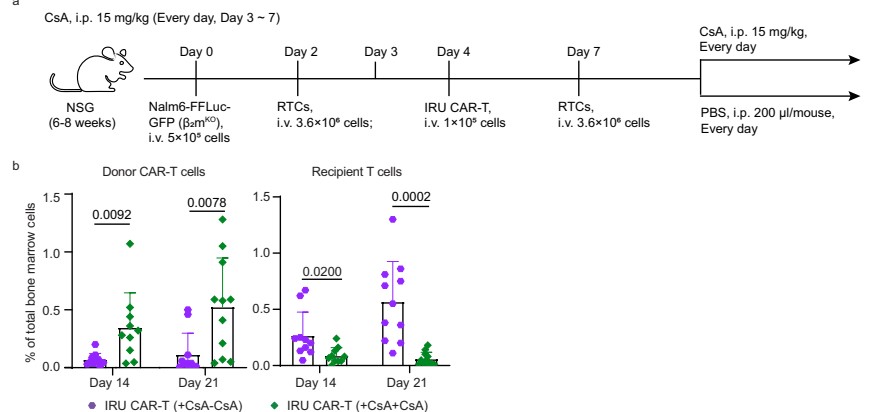

**Fig. 6 | CsA withdrawal restores recipient T cell rejection against IRU CAR-T cells. a** Schematic of ALL mouse model with rechallenged RTCs for (**b**). **b** Frequencies of donor CAR-T (left) and recipient T cells (right) in bone marrow at day 14 and 21 (*n* = 10 and 11 animals in three batches at day 14 and 21 respectively in

each group). All data are means±s.d, *P* values were determined by two-tailed unpaired t test (**b**). IRU CAR-T (+CsA-CsA) as purple circle, IRU CAR-T (+CsA+CsA) as green rhombus.

## Methods

### Study approval
All experiments conducted on human material were approved by the Clinical Research Ethics Committee of the First Affiliated Hospital, Zhejiang University School of Medicine in 2022, No. 1013-Quick. Healthy volunteers were recruited after informed written consent. All animal experiments were conducted in compliance with relevant animal use guidelines and ethical regulations, and approved by Ethics Committee of Hangzhou Normal University No. HSD20210703.

### Cell lines and culture conditions
Nalm6-FFluc-GFP was from Shanghai Model Organisms. Nalm6 -FFluc-GFP-$\beta_2$m$^{KO}$ and Nalm6 -FFluc-GFP-CD19$^{KO}$ was engineered using lenti virus that produced from plasmid lentiCRISPR v2 (addgene)[49,50] containing gRNA sequence 5′-AGTCACATGGTTCACACGGCGTTT-3′, and 5′-AAACCCGAGCAGCGACGTCC-3′, respectively. Jurkat-NFAT-GFP cells were a kind gift from Prof. Wei Chen, Zhejiang University. AAV-293 cells were obtained from the Cell Line Ontology Subset for Chinese National Infrastructure.

Adherent cell lines (AAV-293) and suspension cell lines (Jurkat-NFAT-GFP, Nalm6-FFluc-GFP, and Nalm6 -FFluc-GFP-$\beta_2$m$^{KO}$) were cultured in DMEM or RPMI 1640 supplemented with 10% fetal bovine serum (FBS) respectively at 37 °C in 5% $CO_2$. All cell lines have been routinely tested for *Mycoplasma* and found to be negative.

### Isolation and expansion of human T cells, PBMCs, and activated RTCs
Buffy coats were donated from healthy volunteers. PBMCs were isolated by density gradient centrifugation. T cells were purified using the human T cell enrichment cocktail (STEMCELL RosetteSep), activated with Dynabeads (1:1 beads:cell) Human T-Activator CD3/CD28 (ThermoFisher), and cultured in X-vivo 15 medium (Lonza) supplemented with 10% FBS, 5ng/ml recombinant human interleukin (IL)-7 and IL-15 (BioLegend). The medium was changed to retain the cells at a density

of 0.25-1 × 10$^6$ cells/ml. To obtain activated RTCs, recipient PBMCs were firstly stimulated with irradiated PBMCs from anther donor at a ratio 1:1 for 6 to 7 days in X-vivo 15 medium (Lonza) supplemented with 10% FBS and 20 U/ml IL-2 (Pepro Tech). RTCs were subsequently enriched from the stimulated recipient PBMCs using MojoSort™ Human CD3 T Cell Isolation kit (BioLegend).

### Plasmids construction and AAV production
Plasmids of CAR-T cells were designed based on previous studies[17]. Based on a pAAV-GFP (Addgene) as backbone, we cloned ~400 bp *TRAC* left homology arm (LHA) and right homology arm (RHA) that flanked the gRNA targeting sequences. In the middle of LHR and RHA, a P2A peptide followed by the gene cassette "1928z CAR-T2A-WT/mCNA" was inserted[17,18]. All plasmids were prepared using standard molecular biology techniques and verified using Sanger sequencing.

After these plasmids were constructed, they were transduced into AAV-293 cells along with the helper pDP6 AAV Help/Cap/Rep alternative plasmid (Plasmid Factory) to produce respective AAV6 virus following a standard AAV production protocol and quantified by qPCR reactions[51].

### T cell genetic modification
After initiating T-cell activation for 48 hrs, CD3/CD28 beads were magnetically removed. Fifteen micrograms of Cas9 (homemade) and 8 μg of gRNA (5′-CAGGGUUCUGGAUAUCUGU-3′) (Genscript) were mixed and incubated in 37 °C at least 10 minutes to form ribonucleoprotein (RNP). The RNP was subsequently delivered into 2 × 10$^6$ T cells using Lonza IIB program U-014 in a 0.2 cm cuvette (ThermoFisher). Following the electroporation, cells were diluted into 1 ml culture medium and incubated at 37 °C, 5% $CO_2$ for 1 hr. Afterwards, the cells were harvested by centrifugation at 90g for 10 mins, and recombinant AAV particles were added to the culture at the MOI indicated. Edited CAR-T cells were then cultured as described above and replenished fresh medium to maintain a density of 0.25-1 × 10$^6$ cells/ml every 2 or 3 for 6 days.

To obtain sorted CAR-T cells, CAR-T cells were sorted by Sony cell sorter (SH800). To obtain TCR⁻ T cells, TCR⁺ T cells were removed using MojoSort™ Human CD3 Selection kit (BioLegend).

**Validation assay of mCNA resistance to CsA in Jurkat-NFAT-GFP**
WT/mCNA was expressed on Jurkat-NFAT-GFP cells using lenti virus infection. A total of $1 \times 10^6$ WT/mCNA Jurkat-NFAT-GFP cells were stimulated by Nalm6-FFLuc-GFP for 24 hrs in RPMI 1640 supplemented with 10% FBS in 48-well-plate ($n = 3$ per group), in the presence of 0, 30, 100 and 300 ng/ml CsA. Cells were subsequently harvested and FACS analysis.

**Proliferation assay**
Sorted $1 \times 10^4$ WT/IRU CAR-T (CD3⁻CAR⁺ T cells, the cell subset in Q4 in Fig. 1B) cells were co-cultured with 5-fold irradiated Nalm6 -FFluc-GFP (50 Gy) in U-bottom 96-well tissue culture plates, in X-vivo medium supplemented with 10% FBS. After 48 hrs, CAR-T cells were manually counted every 24 hrs using Trypan Blue ($n = 3$ biological samples).

**Cytotoxicity assays**
To assess the cytotoxicity of CAR-T cells against Nalm6-FFLuc-GFP/ CD19$^{KO}$ Nalm6-FFLuc-GFP cells, sorted IRU/WT CAR-T cells were co-cultured with $1 \times 10^4$ target cells at the indicated E:T ratios in 1 ml X-vivo medium supplemented with 10% FBS in U-bottom deep-well multi well plates at 37°C. Same density of target cells was also cultured alone to determine the maximal luciferase expression (relative light units; RLU) as control ($n = 3$ biological samples). All cells were harvested after 24 hrs and target cells were quantified using the standard protocol of Bright-Lite luciferase assay system (Vazyme). Emitted light was detected in a luminescence plate reader (Molecular Devices iD5). Lysis was determined as $(1- (RLU_{sample})/(RLU_{max})) \times 100$.

**Phenotype assays**
A total of $1 \times 10^5$ sorted CD19/GPC3 WT/IRU CAR-T cells were co-cultured with 5-fold irradiated Nalm6-FFLuc-GFP/HepG2-FFLuc-GFP cells in U-bottom 96-well tissue culture plates in 200 μl X-vivo medium supplemented with 10% FBS for 24 hrs. Cells were harvested and performed FACS. To check the phosphorylation efficiency of WT/IRU CAR-T cells, the cells were intracellular staining by Recombinant Anti-NFAT2 (phospho S237, Abcam, Cat: ab183023, usage dilution 100:1) antibody as the first antibody and Alexa Fluor™ 568 Goat anti-Rabbit IgG (H+L) Highly Cross-Adsorbed Secondary Antibody (Invitrogen, Cat: A11036, usage dilution 200:1) as secondary antibody, following the commercial standard protocol.

**Cytokine assays**
For in vitro cytokine assay, $1 \times 10^5$ sorted CAR-T cells were co-cultured with 5-fold irradiated Nalm6-FFLuc-GFP cells (HepG2-FFLuc-GFP for GPC3 WT/IRU CAR-T cells) in U-bottom 96-well tissue culture plates in 200 μl X-vivo medium supplemented with 10% FBS for 24 hrs. The Golgi Plug protein transport inhibitor brefeldin A (BioLegend) was added for the last 4 hrs of stimulation ($n = 3$ biological samples). For ex vivo cytokine assay, $0.5–5 \times 10^6$ extracted bone marrow cells were stimulated by PMA/Ionomycin for 4 hrs (50 ng/ml, 1μg/ml) in the same condition as above ($n \geq 7$ animals per group).

**Antibodies, extracellular and intracellular staining**
The fluorophore-conjugated antibodies were purchased from Biolegend unless specified. Usage dilutions were 100:1, unless specified. Extracellular staining antibodies include Alexa Fluor 647-Rabbit Anti-Mouse FMC63 scFv Polyclonal Antibody (BIOSWAN, Clone: R19M, Cat: 200102), Brilliant Violet 510-HLA-A2 (Clone: BB7.2, Cat: 343320); Brilliant Violet 510-CD3ε (Clone: OKT3, Cat: 317332); Brilliant Violet 605-CD4 (Clone: SK3, Cat: 344646); Brilliant Violet 711-CD8 (Clone: SK1, Cat: 344734); APC/Cyanine7-CD8 (Clone: SK1, Cat: 344714); APC-CD19

(Clone: HIB19, Cat: 302212) PerCP/Cyanine5.5-CD45RA (Clone: HI100, Cat: 304122); Alexa Fluor 700-CCR7 (Clone: G043H7, Cat: 353244). Intracellular staining antibodies include Pacific Blue-IL-2 (Clone: MQ1-17H12, Cat: 500324); PE-IFNγ (Clone: 4S.B3, Cat: 502509); PerCP/Cyanine5.5-TNFα (Clone: MAb11, Cat: 502926), Alexa Fluor® 647 anti-HA.11 Epitope Tag Antibody (Clone: 16B12, Cat: 682404). Goat anti-Rabbit IgG (H+L) Highly Cross-Adsorbed Secondary Antibody, Alexa Fluor™ 568, Invitrogen™ (Invitrogen, Cat: A11036, usage dilution: 200:1). Recombinant Anti-NFAT2 (phospho S237) antibody (Abcam, Cat: ab183023), DAPI (ThermoFisher, Cat: D1306) and Aqua (Cat: 423102) were used as viability dyes. FcR Blocking Reagent (Equitech Bio Inc.) was used to block Fc receptor staining. Fixation buffer and perm wash buffer were from Biolegend. All extracellular and intracellular staining were performed using an optimized staining protocol from antibody provider company. Samples were acquired on either LSRFortessa (BD Biosciences) or a CytoFLEX and DxFLEX flow cytometer (BECKMAN COULTER) and analysed using FlowJo v.10 software.

**Mixed lymphocyte reaction (MLR)**
Sorted donor CAR-T cells were co-cultured with mismatched HLA-A2 recipient PBMCs or RTCs at the specified ratios in 200 μl X-vivo medium supplemented with 10% FBS in U-bottomed 96-well plates. Prior to coculturing with donor CAR-T cells, recipient PBMC/RTCs were primed by irradiated (30Gy) fresh donor PBMCs at 1:1 in X-vivo medium supplemented with 10% FBS and 50 U/ml of recombinant human IL-2 for 6 days[34]. 5-fold excess of irradiated (50Gy) Nalm6-FFLuc-GFP-β2m$^{KO}$ cells were added into the mixture to stimulate CAR-T cell proliferation. At each indicated time point, total cells from one well were counted by Trypan Blue and stained with anti-human HLA-A2 and CD19 antibodies to distinguish donor and recipient populations ($n = 3$ biological pair samples).

**Mouse strains, breeding condition, tumour monitor, and euthanasia**
NOD.Cg-$Prkdc^{scid}Il2rg^{em1Smoc}$ (NSG) (female, aged 8–12 weeks) were purchased from the Shanghai Model Organisms Center, Inc. (Stock: NM-NSG-001) and bred in the Laboratory Animal Center Hangzhou Normal University (specific pathogen-free level). Other parameters included: dark/light cycle 12/12 (6:00-18:00 light), ambient temperature 21–23°C, and humidity 40–70%. Mice were randomly assigned into control or treatment groups, and bred separately Mouse condition and survival were observed by at least one operator who was blinded to the experiment, in addition to at least one investigator who was not blind to group allocation. All animal experiments were conducted and terminated in compliance with relevant animal use guidelines and ethical regulations, and approved by Ethics Committee of Hangzhou Normal University No. HSD20210703. Mice were euthanised by $CO_2$.

Tumour progression was measured about every 1 week by bioluminescence imaging apparatus (biospace lab photon image optima) after intraperitoneally injecting mice with 150 mg/kg D-luciferin (Gold Biotechnology). The acquisition time was fixed at 1 min for every exposure, and analysis of imaging datasets were used by M3Vision. Briefly, a constant region-of-interest (ROI) was drawn over the tumour region, and the average radiance were measured as total photon/s (ph/s).

**Mouse ALL model**
NSG mice were inoculated with $5 \times 10^5$ Naml6-FFLuc-GFP cells (i.v.) at day 0[17,18], followed by $1 \times 10^5$ CAR-T cells (i.v.) at day 4. CsA (15 mg/kg, i.p.) was started to inject daily from day 3 to 21, and every other day afterwards ($n = 8$ animals per group). For ex vivo assay, mice were euthanised at specified day and their bone marrow cells were harvested from hind limb ($n \geq 7$ animals per group). Bone marrow cells were divided into two portions in specified time point: one was

analysed by FACS using indicated surface antibodies to explore the existence and phenotype of CAR-T cells; the other was stimulated by PMA/Ionomycin for 4 hrs in the presence of brefeldin A, fixed and permeabilized, and performed flow cytometry to detect the cytokine-producing CAR-T cells.

### Mouse ALL model with RTCs

NSG mice were inoculated with Naml6-FFLuc-GFP-β2m$^{KO}$ and CAR-T cells intravenously, and CsA intraperitoneally as above ($n = 8$ animals per group). Specifically, in this model, $3.6 \times 10^6$ HLA-A2 mismatched recipient T cells (RTCs) were injected at day 2 as followed previous research[34]. Of these RTCs, $3 \times 10^6$ T cells were stimulated by CD3/CD28 for 6–8 days, and $6 \times 10^5$ T cells were primed by donor PBMCs as described above. Tumour progression was monitored as above. For ex vivo assay, the bone marrow cells were obtained and analysed as above ($n \geq 7$ animals per group). To establish ALL mouse model with rechallenged RTCs, same dose RTCs were re-injected after 7 days of tumour inoculation. Mice were then divided into two groups after RTCs re-injection: CsA continuous injection (+CsA+CsA) group that injected CsA as before, and CsA removal (+CsA-CsA) group that injected 200 μl PBS intraperitoneally ($n \geq 7$ animals per group).

### Mouse GvHD model

NSG mice were inoculated with $5 \times 10^6$ IRU CAR-T or conventional CAR-T (CAR was transduced by lentivirus. No mCNA sequence was inserted) cells intravenously ($n = 5$ animals per group). Weight and GvHD score were recorded every week. GvHD score was evaluated by parameters including posture, activity, fur, skin and weight loss as previously described[32].

### GvHD in vitro assay

IRU CAR-T cells/conventional CAR-T (made by lentivirus infection) were cocultured with irradiated recipient PBMCs (30 Gy) for 4 days in at 1:1 in X-vivo medium supplemented with 10% FBS and 50 U/ml of recombinant human IL-2 for 4 days. The Golgi Plug protein transport inhibitor brefeldin A (BioLegend) was added for the last 4 hrs of FACS detection.

### RNA-seq and analysis

Sorted WT/IRU CAR-T cells were stimulated with irradiated Naml6-FFLuc-GFP cells for 24 hrs, followed by the enrichment of CD8$^+$ T cells (Miltenyi Biotec) ($n = 3$, 1 biological sample). The purity of CD8$^+$ T cells was confirmed by flow cytometry. The total RNA of $1 \times 10^5$ isolated CD8$^+$ T cells were extracted using Trizol (Invitrogen, Carlsbad) and samples were sent to BGI-Shenzhen following the standard protocol for RNAseq. The fastq files of samples were analysed by STAR (v.2.7.2a) with the reference of GRCh38. The DEGs and enriched pathways were obtained by DEseq2 (v1.34.0) and ClusterProfiler (v4.2.2) respectively. Heatmap of DEGs was visualized by R package pheatmap.

### qPCR analysis

Sorted WT/IRU CAR-T cells were stimulated, CD8$^+$ T cell subset was enriched, and RNA was extracted as described above. cDNA was further obtained using Tiangen RNAprep pure micro kit according to the standard commercial protocol. Primers were synthesized by Tsingke Biotechnology Co.,Ltd., and listed as Table 1 in the Supplementary file. qPCR reactions were set up using ChamQ Universal SYBR qPCR Master Mix according to the standard commercial protocol, and performed in 7500 Real-Time PCR system (Applied Biosystem) along with melting curve. The gene expression was calculated using equation $2^{-\Delta t}$ with the Ct of housekeeping gene β2m as standard.

### Statistics

All in vitro experiments were repeated at least triplicates unless specified. All in vivo experiments were performed with 7 to 24 biological replicates as specified. All experimental data are denoted as mean +/- s.d. unless specified and analysed on GraphPad Prism 10 software. Statistical comparisons between two groups were determined by two-tailed parametric or nonparametric (Mann–Whitney U-test) t-tests for unpaired data or by two-tailed paired t-tests for matched samples. For in vivo experiments, the overall survival comparison was determined by Mantel-Cox log-rank test. $P$ values <0.05 were considered to be statistically significant. The specific statistical test used is stated in the corresponding figure legend.

### Reporting summary

Further information on research design is available in the Nature Portfolio Reporting Summary linked to this article.

## Data availability

The RNA-seq data has been deposited in the Sequence Read Archive (SRA) under accession code PRJNA880631. Source data are provided with this paper.

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

## Acknowledgements

This project was funded by National Natural Science Foundation of China grants, 82161138028 (H.X), 81870136 (H.X), 82170141 (H.X), 31971324 (J.S.), 82161138028 (J.S.), 31870899 (X.Z), and 32070899 (X.Z.). We thank Prof. Wei Chen and Zeng laboratory members for helpful discussions and necessary assistance. We thank Yanwei Li and Yueting Xing from the Core Facilities, Zhejiang University School of Medicine for their technical support.

## Author contributions

Y.Z., H.F., and G.W. designed the study, performed the experiments, analysed and interpreted the data, and wrote the manuscript. G.Y. assisted with the in vivo experiments and analysed the RNA-seq data. R.D., J.L., Y.L., and Y.W. assisted in the vector construction, in vitro, and ex vivo experiments. P.L. helped to analyse the RNA-seq data. C.Z. and W.Y. purified the Cas9 protein. H.X. and J.S. analysed and interpreted the data, and wrote the manuscript. X.Z. designed the study, analysed and interpreted the data, wrote the manuscript, and took the main responsibility for the whole project. All authors give their consent to publishing this work.

## Competing interests

A patent application has been submitted based in part on the results presented in the manuscript. Y.Z., H.F., G.W., and X.Z. are listed as the inventors. The remaining authors declare no competing interests.
