## [Peer Review File · Nature Communications]

REVIEWER COMMENTS

Reviewer #1 (expert in CAR T cells and haematological malignancies):

The authors engineered CAR T cells expressing a mutated calcineurin subunit A (CNA) and a CD19-specific CAR into the T-cell receptor constant (TRAC) locus in order to develop CAR T cells resistant to a common immunosuppressant (cyclosporine A) and with a potential off-the-shelf use. They show that these cells they called IRU CAR-T cells displayed better effector function in vitro in vivo in the presence of cyclosporine A, compared to the CAR-T cells with wild type CNA.

This work is important to address a major gap in the field of CAR T-cell therapy given that allogeneic CAR T-cell therapies are still at their infancy and many concerns impair their application in clinical trials.

However, it is unclear whether the differences observed in terms of phenotype in vitro and in vivo are specific to CAR T-cells or mainly depend on a CSA-resistance mechanism.

Lack of validation of the transcriptomic results diminishes enthusiasm and a proper validation by qPCR or WB of the DEG could provide insights into the molecular mechanisms underpinning promising observations.

In vivo it is unclear whether IRU CAR T-cells seem performing better under CSA simply because control die earlier and thus, additional controls would be required.

The central memory phenotype which is attractive is not well characterized and the mechanisms underpinning this aspect of the study are understudied.

Finally, while CSA is a commonly used immunosuppressant, it is not the only one and it may be worth to show that this approach is valid in different contexts as well.

Reviewer #2 (expert in CAR T cells and adoptive cell therapy):

The authors present a platform for developing an allogeneic CAR T cell, aiming to minimize GVHD and rejection by inserting the transgene into the TRAC locus and adding resistance to CsA, respectively. Some comments for the authors:

- Did the authors perform prior work on CAR T cell insertion into TRAC? There are no references in the manuscript, and no data comparing regular CAR T with TRAC-inserted CAR T throughout the paper. The same reasons for comparing CsA-resistant vs non resistant cells for equivalence holds for TRAC-inserted and regular CAR T
- Did the authors test whether their CAR is only specific to CD19? For example, it is unable to recognize CD19-negative cell lines?
- Is the approximately ~80% knockdown of TCR representative of many populations
- Can the authors clarify the reasoning behind their selection of the CsA dosage? The cited paper seems to study 200 ng/mL as the cutoff for relationship with GVHD, not 300, and they were initially targeting 250 ng/mL?
- Can the authors clarify why PMA/ionomycin was used throughout the manuscript to stimulate T cells in vitro instead of through their CD19 CAR? It does not test CAR T cell activity, and the results will not be as relevant to the goals of showing that CARs with the CsA mutation or insertion via TRAC are equally efficacious.
- Why was there a P2A cleavage signal prior to the CD19CAR? What gene precedes the CAR?
- Why were irradiated Nalm6 needed for measuring MLR? This is unclear.
- For Figure 1, as mentioned in one of the bullet points above, comparisons with a non TRAC CAR would be useful. Are MFIs similar, for example? Can the authors comment on the maximum lysis of NALM-6 seen in Fig1D and how this relates to similar CAR s in the literature? It would be interesting to see whether the TRAC-inserted CAR has a slightly lower upper limit, making further decreases due to CsA difficult to quantify. Are cells in Fig 1F gated on just the CAR T cells (the ~80% in the population)?
- For Figure 2, are the values for WT + vs -CsA and IRU + vs -CsA statistically significant in 2D? The authors only analyzed WT vs IRU and do not show whether CsA has an actual effect within

groups.

- For Figure 3, do the authors have a Day 0 image of the co culture? Also, it's curious to see that CsA does not eliminate the recipient T cells completely/or as much as the WT CAR T. Can the authors comment on why this would be the case? Finally, it is worth noting too that donor T cell allo reactivity should also exist against recipient T cells. From the co culture experiments, this does not seem to be observed?
- For Figure 4, why is the IRU CAR-T (-CsA) group dropped from the comparisons?

Reviewer #3 (expert in T cell signalling):

This paper deals with strategies to avoid GVH and HVG disease when using CAR T cells. The authors used CRISPR/Cas9 to knock the calcineurin A chain (CNA), either WT or a mutant that can't bind to cyclosporin A/cyclophilin A plus a CAR, into the TCR α locus. This accomplishes three things in successfully mutated cells: endogenous TCR expression is lost, CAR expression is gained, and the cells overexpressing the mutated CNA are resistant to the immunosuppressive effects of cyclosporin A. They show that such human primary T cells transferred to tumor-bearing NSG mice can persist and be effective in the presence of cyclosporin A, and can be eliminated when the mice have allogenic T cells and administration of cyclosporin A is discontinued. Overall, the data are consistent and support the potential utility of such an approach. The paper, however, is poorly written, inadequately documented, and the explanations of the results and discussion incomplete or missing.

1. In Fig. 1J, many genes are lower in IRU CAR-T vs WT CAR-7 in the absence of CsA. Why? CNA is overexpressed in the transduced cells (the endogenous gene isn't silenced), so is it possible that the IRU mutation is affecting calcineurin functions other than just binding to CyPA? Does it then act as a dominant negative? Some attempt should be made to confirm the RNA-seq profile (i.e. qPCR for message and determination of protein for some genes, like granzyme B). This certainly warrants a discussion.

2. In Supp. Fig.2F, how were the averages calculated? They don't reflect the data. For example, in the first bar for IL-2 the 3 blue dots are all below the average. There are other examples in this figure.

3. In Fig. 3C the decrease in Donor CAR-T cells is seen only at one time point, 9 days. The experiment should show at least one more time point, such as 12 days, to confirm the loss of CAR-T. Also, a control without RTC under the conditions, which differ a bit from 1E, should be shown.

4. In Fig. 2D at 14 days (and 21 days to a lesser extent) it appears that the % of IRU CAR-T cells in the bone marrow is actually greater in the presence of CsA. And in Extended Data 2C they are elevated at 10 days (although the error bars are large). Is that because CsA has an effect on the numbers of endogenous murine bone marrow cells? Did their number decrease?

5. Throughout the P values in figure bar graphs are shown comparing WT to IRU CAR-T (see 1F, 2D as examples). What's more important to show, though, is the intra-group comparison of (-CSA) vs. (+CSA). It would also make it easier to compare the behavior of the two groups of CAR-Ts if they were shown in the figure in this order (left to right): WT CAR-T (-CSA), WT CAR-T (+CSA), IRU CAR-T (-CSA), IRU CAR-T (+CSA).

6. The statement on line 185-186 that "the cytotoxic functions of the detectable residual WT CAR-T (+CSA) cells were significantly impaired as quite low frequencies of cytokine-secreting cells were detected..." is misleading. As noted, they measured cytokine production, not CTL activity.

7. Fig. 3 is inadequately explained and very difficult to follow. Because they want to set up an allogenic MLR it seemed at first that this was being done by mismatching HLA-A2, but that made no sense because it was the donors that were A2-. It seems, rather, than A2 was just being used as a marker so that the donor and recipient cells could be distinguished by flow. That should be made clear. On line 198 it is written "IRU CAR-T (+CSA) group showed significantly higher CAR-T

percentage" -- compared to what? I imagine they mean the WT CAR-T (+CSA) group. But the percentages are changing in the other groups too, because the RTCs in the absence of CSA are expanding more than the CAR-T. This is not straightforward because in 3 of the groups both the donor CAR-T cells and the RTC are cyclosporin A-sensitive. This confusing experiment requires more discussion.

8. How was the percent of CAR-T cells that were double producers, such as in Fig. 4G and H, calculated? In G for IRU CAR-T (+CSA) the % of TNF+ is about 25% and IFN+ maybe 8%, yet the TNF+IFN+ groups is perhaps 15%. The maximum it could be is 8%, and that's if every IFN+ cell also produced TNF.

9. It's not clear how the cytotoxicity assay in Fig. 1D was performed. According to the Methods, line 402, "sorted" cells were used. Sorted on what? The CAR? CD8+?

10 Fig. 4F and supplemental 4D is incomprehensible. What do the numbers mean (percent?). What do the colors mean? What does the heat scale mean?

11. In Fig. 4B why was analysis of IRU CAR-T (-CSA) not shown? It would seem to be an important control that presumably would show that the IRU CAR-T cells were just as susceptible to WT CAR-T cells to rejection.

Minor point:

1. Extended Data 2C. The symbols in the key for the top panel (% of total bone marrow cells) don't match those in the plot.

RESPONSE TO REVIEWERS' COMMENTS

We would like to appreciate reviewers for your evaluations and valuable comments. We have intensively revised our manuscript based on these comments and have responded to the reviewers' comments point-by-point below. The comments from the reviewers are shown in black *italic bold* and our responses are shown in black regular. We have shown the updated or adjusted figures in the Response Figures (Fig. RX) attached below and highlighted the related context, meanwhile also labeled their new citations in the revised manuscript.

REVIEWER 1:

The authors engineered CAR T cells expressing a mutated calcineurin subunit A (CNA) and a CD19-specific CAR into the T-cell receptor α constant (TRAC) locus in order to develop CAR T cells resistant to a common immunosuppressant (cyclosporine A) and with a potential off-the-shelf use. They show that these cells they called IRU CAR-T cells displayed better effector function in vitro in vivo in the presence of cyclosporine A, compared to the CAR-T cells with wild type CNA.

This work is important to address a major gap in the field of CAR T-cell therapy given that allogeneic CAR T-cell therapies are still at their infancy and many concerns impair their application in clinical trials.

1. However, it is unclear whether the differences observed in terms of phenotype in vitro and in vivo are specific to CAR T-cells or mainly depend on a CSA-resistance mechanism.

We thank Reviewer 1 for raising the questions about the phenotype of IRU-CAR-T cell depend on either CAR-T cell specific or CSA-resistance. To address this issue, we generated a GPC3 specific CAR-T (GPC3-CAR-T) and a CD19 specific CAR-T (CD19-CAR-T, i.e. WT/IRU CAR-T in this paper) with WT CNA or mCNA from the same donor, and compared the difference of phenotype and cytokine production between them. These CAR-T cells were stimulated with either HepG2-FFluc-GFP (GPC3⁺) or Nalm6-FFLuc-GFP cells (CD19⁺) for 24hrs, respectively, and performed

FACS to check phenotypes (Fig. R1, Extended data Fig. 1I-K). In GPC3-CAR-T cells, the frequency of T_{EM} was significantly different between WT CAR-T and IRU CAR-T in the presence of CsA. In CD19-CAR-T cells, the frequency of T_{EM} and T_{EMRA} was significantly different between WT CAR-T and IRU CAR-T in the absence of CsA, respectively (Fig. R1A, Extended data Fig. 1I). Comparing the phenotype of GPC3-CAR-T and CD19-CAR-T in the absence of CsA (Fig. R1C, Extended data Fig. 1K), we found CD19-CAR-T cells had higher frequency of T_{EM} than that in GPC3-CAR-T cells regardless of CNA or mCNA expression.

In the perspective of cytokine, the trend of cytokine production of GPC3-CAR-T cells was the same as CD19-CAR-T cells with/without CsA (Fig. R1B, Extended data Fig. 1J), i.e., the addition of CsA would significantly abolish the cytokine production of WT GPC3 CAR-T cells, but not IRU GPC3 CAR-T cells. However, the detailed frequency of cytokine-secreting CAR-T cells were not the same (Fig. R1C, Extended data Fig. 1K) in the absence of CsA for different CAR-T cells. These data suggested that different CAR-T cells may affect phenotype. However, CsA resistant mechanism drove the similar pattern of cytokine production.

Fig. R1: Different CAR-T may have different phenotypes, however, have the similar cytokine production pattern. (A) The phenotype of CD19/GPC3 WT/IRU CAR-T cells with/without CsA. (B) The cytokine of GPC3 WT/IRU CAR-T cells with/without CsA. (C) The phenotype and cytokine comparison between CD19/GPC3 WT/IRU CAR-T cells in the absence of CsA.

2. Lack of validation of the transcriptomic results diminishes enthusiasm and a proper validation by qPCR or WB of the DEG could provide insights into the molecular mechanisms underpinning promising observations.

Many thanks for reminding the importance of hands-on experiments to validate bioinformatic data. Following the same protocol, CD8⁺ WT/IRU CAR-T (+/- CsA) cells were obtained, and their cDNA were extracted. qPCR assays were performed (Fig. R2, added in manuscript as Fig. 1K), and the gene expression were calculated using equation $2^{-\Delta t}$ with the Ct of housekeeping gene $\beta 2m$ as standard. As shown, qPCR followed the similar trend of RNA-seq, which further confirmed the accuracy of RNA-seq.

Fig. R2. qPCR validation of representative genes in bulk RNA-seq.

From the qPCR validation assay, we found that the trend of genes expression level, such as the NFAT-regulated activation, survival and proliferation (e.g. *Il2*¹, *Il3*¹, *Il5*², *PCNA*^{3,4}, *CSF2*⁵, *CDC25A*⁶, *CDK4*^{7,8}) and NFAT-regulated cytotoxicity (e.g. *Ifng*¹, *Gzmb*¹, and *Tnf*¹) were similar in that of RNA-Seq data. Generally, genes that were positively related to proliferation were highest in WT (-CsA) group, and gradually decreased in IRU (-CsA). The addition of CsA significantly diminished the expression of such genes in WT CAR-T groups, but not in IRU CAR-T groups. In contrast, the apoptosis-related genes (e.g. , *ATM*⁹ and *NTRK1*¹⁰) were up-regulated in WT (+CsA) group, rather than any other three groups. These results consistently showed that IRU CAR-T cells could resist the CsA and retain their effector function in the perspective of RNA level.

In addition, as RNA-seq and qPCR showed, many genes that positively related to NFAT activation were higher in WT CAR-T than that in IRU CAR-T in the absence of CsA. This raised a very important question: could mCNA affect T cell effector functions differently compared to WT CNA? (added as Extended data Fig. 2) To address this issue, we firstly expressed WT CAR-T and IRU-CAR-T in Jurkat-NFAT-GFP reporter cell line, then stimulated these cells with Nalm6 cell line without CsA to compare the GFP expression level. We found the frequency of GFP⁺ cells in WT CAR-T-Jurkat-NFAT-GFP cell line were higher than that in IRU-CAR-T-Jurkat-NFAT-GFP cell line in the absence of CsA, when stimulated by Nalm6 (Fig. R3A; added as Extended data Fig. 2A). We hypothesized that this phenomenon is due to the less efficient dephosphorylation of NFAT in mCNA compared to WT CNA. To confirm this, we applied phosphor-flow to detect the level of NFAT phosphorylation in primary T cells. Upon the stimulation of Nalm6 cell line, we also found that the MFI of the phosphorylated NFATc1 in WT CAR-T cells was lower than that in IRU- CAR-T cells without CsA, but the MFI of the phosphorylated NFATc1 in WT CAR-T cells was higher than in IRU- CAR-T cells with CsA (Fig. R3B, C; added as Extended data Fig. 2B, C). Collectively, these data indicated that mCNA had less efficient for NFAT dephosphorylation which affected downstream of NFAT signaling pathway and T cell effector functions.

Fig. R3: WT and mCNA have different deficiencies of dephosphorylation of NFAT signal. **A**. GFP⁺ WT CAR-T-Jurkat-NFAT-GFP cells were significantly higher than GFP⁺ IRU CAR-T-Jurkat-NFAT-GFP cells, upon the stimulation of Nalm6. **B,C**, representative histogram (**B**) and bar graph of MFI (**C**) of the phosphorylated NFATc1 expression in primary WT/IRU CAR-T cells.

3. *In vivo it is unclear whether IRU CAR T-cells seem performing better under CSA simply because control die earlier and thus, additional controls would be required.*

Thanks for pointing that it might not be satisfactory to observe that IRU CAR-T cells performed better due to the delayed death. For a more convincing conclusion, we added IRU (-CsA) control group. The experiment was designed to exclude the IRU (-CsA) group in previous version, as the novel universal CAR-T was expected to use in the setting with CsA, but not without CsA, in the presence of recipient immune system. However, after careful consideration, we accepted that the IRU (-CsA) control group is necessary to be added in a more scientific perspective.

Fig. R4 (all added in Fig. 4, Extended Data Fig. 6) shows the tumor burden and survival of the IRU (-CsA) group in the mouse model in Fig. 4. Other relative data have been added in the corresponding figures in Fig. 4 and Extended Data Fig. 6. In these updated data, IRU CAR-T (-CsA) group showed a similar trend of WT CAR-T (-CsA) group.

Fig. R4: IRU CAR-T could not retain anti-leukemia functions in the presence of RTCs *in vivo* without CsA. **A.** Tumor burden (average radiance, ph/s) in IRU CAR-T (-CsA) (n=5 mice per group). **B.** Kaplan-Meier analysis of the mice survival in mice treated with WT CAR-T cells (+/-CsA) and IRU CAR-T cells (+/-CsA) (n=5 mice per group).

In addition, to confirm the comparability and consistency of the model with previous data, 3 additional mice in WT (+/- CsA) and IRU (+CsA) groups were performed simultaneously as internal reference (green line in each graph, No. 6, 7, and 8). As shown, this bench generally followed the similar trend in previous version, and IRU (-CsA) group showed similar trend like WT (-CsA) group (Fig. R5A).

Fig. R5. Updated data in mouse ALL with recipient T cells model. **A**, Tumor burden (average radiance, ph/s) in WT CAR-T cells (+/-CsA) and IRU CAR-T cells (+/CsA) were performed along with Fig. R5 as internal reference (green line in each graph, No. 6, 7, and 8). **B**, Kaplan-Meier analysis of the mice survival in each experimental group (n=5 mice in IRU WT CAR-T (-CsA) group and n=8 in the rest of groups)

In our previous figure, there is a mouse in WT (-CsA) group survived much better than rest of mice in the same group, raising the concern whether the survival difference between WT (-CsA) and IRU (+CsA) groups was due to limited mice number. It is worth noting that the mouse survived above 50 days in previous version did not happen in the new bench (both in WT and IRU groups without CsA, Fig. R5B), indicating the one survived above 50 days might be an outlier. This result showed that the maximum survival time of WT/IRU CAR-T (-CsA) was below 50 days, which had significantly difference with IRU CAR-T (+CsA) group. We hope these additional controls add more confidence to support our conclusion.

4. The central memory phenotype which is attractive is not well characterized and the mechanisms underpinning this aspect of the study are understudied.

Thanks for helping this therapy going deeper in a perspective of molecular mechanism. In our data, we showed that WT-CAR-T (+CsA) had less T_{CM} cells at day14 in extended data Fig. 2D, and IRU-CAR-T (-CsA) and IRU-CAR-T (+CsA) had more T_{CM} cells at day 24 in extended data Fig. 4E. The underline mechanism of this phenomenon was unclear.

The level of NFAT dephosphorylation reflected the TCR signaling strength in T cells when they were activated. TCR signalling strength contributes to T cell differentiation: the higher TCR signalling strength is associated with the more effector phenotype, and the lower TCR signalling strength is prefer to display the more memory phenotype¹¹. As mentioned above in point 2, due to the lower dephosphorylation efficiency of NFAT in mCNA compared with WT CNA, when CAR-T cells were activated, the signal strength of NFAT in different groups ranged from high to low as follows: WT-CAR-T (-CsA), IRU-CAR-T (-CsA), IRU-CAR-T (+CsA) and WT-CAR-T (+CsA). Therefore, we hypothesized that WT-CAR-T (-CsA) had highest effector phenotype and WT-CAR-T (+CsA) had highest memory phenotype. To examine it, we used three gene set from Gene set enrichment analysis (GESA), GSE23321_central_vs_effector_memory_CD8_T_cell_DN/UP, GOLDRATH_EFF_VS_MEMORY_CD8_TCELL_DN/UP, and GSE9650_EFFECTOR_VS_MEMORY_CD8_TCELL_DN/UP. to analyze our RNAseq data in Fig. 1 (Fig. R6 A-F). We found there was no difference between IRU-CAR-T (-CsA) group and IRU-CAR-T (+CsA) group. However, both IRU-CAR-T (-CsA) and IRU-CAR-T (+CsA) groups expressed more central memory and less effector related genes compared to WT-CAR-T (-CsA) group, suggesting that the gene expression profiles of IRU-CAR-T cells favoured a central memory phenotype compared to WT-CAR-T cells. Consistently, when NFAT signalling was completely blocked, i.e. the addition of CsA, the NFAT-associated effector function genes were significantly downregulated, resulting in reduced expression of effector-related genes

in the WT-CAR-T (+CsA) group, which might contribute a “T_{CM}-like” phenotype *in vitro*. This observation suggested that NFAT signalling strength indeed affected the effector and memory phenotype of CAR-T cells.

To further confirm this hypothesis, we investigated the expression level of effector and memory-related genes by qPCR (Fig. R6G). We found that the expression level of *EOMES*¹², *IL7R*¹³, the essential genes for central memory development, was downregulated in WT-CAR-T groups compared to IRU-CAR-T groups. Previous study showed that the deficiency of NFATc1 led to the increase of EOMES, replying the association between lower NFATc1 level and higher EOMES expression¹. This also further confirmed the speculation above that the lower NFATc1 dephosphorylation of IRU CAR-T might contribute to the preference of T_{CM} phenotype. In contrast, *TBX21*, the critical gene for effector function for CD8⁺ T cells, was downregulated in WT-CAR-T (+CsA) groups compared to rest of groups. In addition, the expression level of plenty of downstream genes and proteins of NFAT signalling pathway was consistent with the NFAT signalling strength among different groups as mentioned above in point 2 (e.g. *IL2*, *IFNG*, and *TNF*, Fig. 1F, R2). Finally, as WT-CAR-T (+CsA) group deficient of IL-2 expression and proliferation, the less T_{CM} phenotype of WT-CAR-T (+CsA) group were observed *in vivo*. Taken together, these data suggested a possible mechanism of IRU-CAR-T cells has more memory phenotype related genes expression compared to WT-CAR-T cells.

Fig. R6: RNA-seq genes showed the similarities and differences between IRU CAR-T (-CsA) and WT CAR-T (-CsA) (**A**); IRU CAR-T (+CsA) and WT CAR-T (+CsA) (**B**); WT CAR-T (+/-CsA) (**C**); IRU CAR-T (-CsA) and WT CAR-T (+CsA) (**D**); IRU CAR-T (+CsA) and WT CAR-T (-CsA) (**E**); IRU CAR-T (+/-CsA) (**F**) using gene set GSE23321_central_vs_effector memory_CD8_T cell_DN/UP, GOLDRATH_EFF_VS_MEMORY_CD8_TCELL_DN/UP, and GSE9650_EFFECTOR_VS_MEMORY_CD8_TCELL_DN/UP (**G**). qPCR validation of memory and effect related genes TBX21, IL7R, and Eomes.

5. Finally, while CSA is a commonly used immunosuppressant, it is not the only one and it may be worth to show that this approach is valid in different contexts as well.

Thanks for helping us broaden the usage of IRU CAR-T cells. To further test whether IRU CAR-T cells are applied in other contexts, we checked the activation of NFAT signalling pathway of WT/IRU CAR-Jurkat-NFAT-GFP cells in the absence/presence of another important immunosuppressant tacrolimus (FK506) in a clinical blood concentration range¹⁴ (Fig. R7). Unfortunately, IRU CAR-T cells failed to show the resistance to FK506 like that to CsA. The NFAT signalling (representing by the expression of GFP) of IRU CAR-T cells showed dose-dependent decline, indicating current IRU CAR-T cells can only be applied in the context of CsA.

Fig. R7: WT/IRU Jurkat-NFAT-GFP cells were stimulated with Nalm6 in the dose-escalated FK506.

- 1 Klein-Hessling, S. *et al.* NFATc1 controls the cytotoxicity of CD8(+) T cells. *Nat Commun* **8**, 511, doi:10.1038/s41467-017-00612-6 (2017).
- 2 Chong, L. K. *et al.* Proliferation and interleukin 5 production by CD8hi CD57+ T cells. *Eur J Immunol* **38**, 995-1000, doi:10.1002/eji.200737687 (2008).
- 3 Kurki, P., Lotz, M., Ogata, K. & Tan, E. M. Proliferating cell nuclear antigen (PCNA)/cyclin in activated human T lymphocytes. *J Immunol* **138**, 4114-4120 (1987).
- 4 Simonett, S. P. *et al.* Identification of direct transcriptional targets of NFATC2 that promote beta cell proliferation. *J Clin Invest* **131**, doi:10.1172/JCI144833 (2021).
- 5 Shi, Y. *et al.* Granulocyte-macrophage colony-stimulating factor (GM-CSF) and T-cell responses: what we do and don't know. *Cell Res* **16**, 126-133, doi:10.1038/sj.cr.7310017 (2006).
- 6 Kittipatarin, C., Li, W., Durum, S. K. & Khaled, A. R. Cdc25A-driven proliferation regulates CD62L levels and lymphocyte movement in response to interleukin-7. *Exp Hematol* **38**, 1143-1156, doi:10.1016/j.exphem.2010.08.010 (2010).
- 7 Romero-Pozuelo, J., Figlia, G., Kaya, O., Martin-Villalba, A. & Teleanu, A. A. Cdk4 and Cdk6 Couple the Cell-Cycle Machinery to Cell Growth via mTORC1. *Cell Rep* **31**, 107504, doi:10.1016/j.celrep.2020.03.068 (2020).
- 8 Shen, T. & Huang, S. The role of Cdc25A in the regulation of cell proliferation and apoptosis. *Anticancer Agents Med Chem* **12**, 631-639, doi:10.2174/187152012800617678 (2012).
- 9 Shiloh, Y. ATM and related protein kinases: safeguarding genome integrity. *Nat Rev Cancer* **3**, 155-168, doi:10.1038/nrc1011 (2003).
- 10 Dadakhujaev, S. *et al.* Interplay between autophagy and apoptosis in TrkA-induced cell death. *Autophagy* **5**, 103-105, doi:10.4161/auto.5.1.7276 (2009).
- 11 Daniels, M. A. & Teixeira, E. TCR Signaling in T Cell Memory. *Front Immunol* **6**, 617, doi:10.3389/fimmu.2015.00617 (2015).
- 12 Kaech, S. M. & Cui, W. Transcriptional control of effector and memory CD8+ T cell differentiation. *Nat Rev Immunol* **12**, 749-761, doi:10.1038/nri3307 (2012).
- 13 Bradley, L. M., Haynes, L. & Swain, S. L. IL-7: maintaining T-cell memory and achieving homeostasis. *Trends Immunol* **26**, 172-176, doi:10.1016/j.it.2005.01.004 (2005).
- 14 Li, Z. *et al.* Tacrolimus Induces Insulin Resistance and Increases the Glucose Absorption in the Jejunum: A Potential Mechanism of the Diabetogenic Effects. *PLoS One* **10**, e0143405, doi:10.1371/journal.pone.0143405 (2015).

Reviewer 2

The authors present a platform for developing an allogeneic CAR T cell, aiming to minimize GVHD and rejection by inserting the transgene into the TRAC locus and adding resistance to CsA, respectively. Some comments for the authors:

1. Did the authors perform prior work on CAR T cell insertion into TRAC? There are no references in the manuscript, and no data comparing regular CAR T with TRAC-inserted CAR T throughout the paper. The same reasons for comparing CsA-resistant vs non resistant cells for equivalence holds for TRAC-inserted and regular CAR T

Thanks for reminding the importance of comparison with previous researches, which will definitely make current research more convincing and significant. We regret that the design of CAR-T cell insertion into TRAC has not been clearly cited and poorly described in previous version.

As far as we know, Justin et. al firstly developed the CAR-T cell insertion into TRAC in 2017¹. By comparing regular CAR-T, they illustrated that the advantages of this CAR-T are better persistence and better tumor control. One of our authors, Jie Sun, has also used such design in 2019². More lines (lines 122-125) have been added and highlighted to further describe the TRAC design. In addition, our goal is to construct a universal CAR-T cell platform, which required to address GvHD and HvGR. For this approach, AAV based knock-in technique combined TCR KO and CAR expression in a single step which facilitated CAR-T cell preparation.

However, more experiments regarding the comparison between regular CAR-T (transduced by lentivirus, LV CAR-T) and our CAR-T (AAV CAR-T) were conducted in our lab, to further confirm the methodology of AAV knock-in technique for producing CAR-T cells. The comparison of killing assay (Fig. R1A) and MFI (Fig. R1B) of the two CARs were compared. As shown, the MFI of LV CAR-T cells were 2.5-fold less than that of our CAR-T cells, and the killing rate were slightly lower than that of AAV CAR-T, but showing a similar trend.

Fig. R1: Killing assay (A) and MFI (B) comparison between LV/AAV WT/IRU CAR-T cells.

Regarding the question about whether regular CAR-T with mCNA can also resist CsA, our data showed that the resistance of CsA depended on the mCNA sequence, rather than the method of manufacturing CAR-T cells. In Fig. R2A, WT and IRU CAR was expressed on Jurkat-NFAT-GFP using lenti-virus. The GFP expression (represented the activation of T cells from NFAT signaling) of WT- Jurkat-NFAT-GFP cell line was inhibited by CsA, while that of IRU- Jurkat-NFAT-GFP cell line was not. Also, in Fig. R2B, WT/IRU CAR was transduced into primary T cells by lenti virus. Similar trend that only WT CAR-T cells produced significantly decreased IFN γ was observed, consistently indicating that both regular CAR-T and our AAV CAR-T could maintain T cell effect with mCNA.

Fig. R2: WT/IRU was expressed in Jurkat-NFAT-GFP cells (A) and primary T cells (B). (A). Normalize GFP expression (relative to the GFP expression without CsA) showed that IRU Jurkat-NFAT-GFP cells could resist CsA, while WT could not. (B). The secretion of IFN γ was significantly inhibited by CsA in WT CAR-T cells, but not in IRU CAR-T cells.

2. Did the authors test whether their CAR is only specific to CD19? For example, it is unable to recognize CD19-negative cell lines?

The scFvs of CD19 CAR we used derived from the murine FMC63, which has been published in clinical trials³. However, as a standard research pipeline, it is regret to show the specificity validation of CD19 in present research. To allay the concerns whether the CAR-T cells are specific targeting to CD19 and tumor cells, the cytotoxicity of WT/IRU CAR-T cells were re-validated in Nalm6-CD19^{KO} cell line (the grey line with triangle up/down symbols, Fig. R3, replacing Fig. 1D). As shown, WT/IRU CAR-T could not kill Nalm6-CD19^{KO} with the increasing of E/T ratio. This result showed the incapability of CAR-T cells when the tumor target is missing, indicating the specificity of CAR-T cells against tumor target. This result was the consistent with previous researches regarding the specificity of the CD19 CAR¹.

Fig. R3: Cytotoxic activity using Nalm6-FFluc-GFP or CD19^{KO} Nalm6-FFluc-GFP as target cells (n=3, 1 biological sample) in 24 hrs cytotoxicity assay at different E:T ratios. Target cell lysis rate was calculated by $(1 - (RLU_{sample}) / (RLU_{max})) \times 100$ (RLU: relative luminescence units).

3. Is the approximately ~80% knockdown of TCR representative of many populations

Yes, ~80% knockdown of TCR represented many populations. More knockout of TCR FACS were shown in fig. R4

Fig. R4: Representative flow plots to show the knockout of TCR in different donors.

4. Can the authors clarify the reasoning behind their selection of the CsA dosage? The cited paper seems to study 200 ng/mL as the cutoff for relationship with GVHD, not 300, and they were initially targeting 250 ng/mL?

Thanks for helping us double check the CsA dosage. The cited paper⁴ reported the “median concentration of CsA in the blood 1st, 2nd, 3rd, and 4th week after allo-HSCT were 218, 235, 263, and 270 ng/ml”. Along this line, 300 ng/ml should be an effective and frequently observed CsA blood concentration, which has been consulted and double checked with our clinic collaborators. Further, some other researches also showed that the effective CsA concentration is approximately in this range (e.g. 200-400 ng/ml⁵). So, it is reasonable to test IRU CAR-T cells in 300 ng/ml. In addition, different doses of CsA have been tested in extended figure 1A (Fig.R2A), which showed that the effect function of IRU CAR-T could be retained in a concentration range.

5. Can the authors clarify why PMA/ionomycin was used throughout the manuscript to stimulate T cells in vitro instead of through their CD19 CAR? It does not test CAR T cell activity, and the results will not be as relevant to the goals of showing that CARs with the CsA mutation or insertion via TRAC are equally efficacious.

In previous version, we used PMA/ionomycin to stimulate CAR-T cells in extended data Fig. 1A and all *ex vivo* experiments (Fig. 2D-F and 4F-G.) We used Nalm6 cell line to stimulate CAR-T cells in Fig. 1D-J, Fig. 3, extended data Fig. 1E-H and extended data Fig. 3. We are sorry that the stimulation way might not be clearly highlighted We have highlighted all the stimulation way in revised manuscript.

We regret to use PMA/ionomycin to stimulate Jurkat-NFAT-GFP (previous Extended Data Fig. 1A), as we thought the preliminary experiment was aimed at validating the mutation on CnA locus and there is no need to stimulate T cells from CAR. However, after considering your advice, we agreed that CAR signal should be also evaluated in this system, which is more straightforward and convincing. Therefore, this experiment was re-performed. WT/IRU Jurkat-NFAT-GFP using Nalm6 to stimulate in the dose-escalated CsA (Fig. R2A, replacing Extended Data Fig. 1A in previous version). Fig R2A used normalized GFP expression, the relative frequency of GFP⁺ cells of different CAR-T cells in the dose-escalated CsA treatment compared to no CsA treatment. Consistent results were shown: the NFAT signaling of WT CAR-T cells were inhibited by CsA, while that of IRU CAR-T cells were not.

In addition, the PMA/ionomycin usage in *ex vivo* experiments (Fig. 2E-F and Fig. 4G-H) may not be avoided. The percentage of CAR-T cells were extremely low (generally below 1%, even 0.5%). This will result in at least two major problems. 1. The stimulation of CAR-T cells might be not sufficient as the other cells might overwhelmed CAR-T cells; 2. The detection of CAR-T cells may be impossible if 5-fold of Nalm6 were added for stimulation. For example, 5×10^6 Nalm6 is required to stimulate 1×10^6 bone marrow cells, to ensure the CAR-T cells were contacted with tumor cells. Then, the percentage of CAR-T cells would become diluted to 0.16%

(assuming CAR-T% in bone marrow=1% before stimulation) in this stage, not to mention that stimulation might result in the death of CAR-T cells and contribute a further lower CAR-T percentage in reality. This will be poorly operational and might lead to the low inaccuracy of data. This probably why many researches used PMA/ionomycin to stimulate CAR-T cells in *ex vivo* assays⁶.

6. Why was there a P2A cleavage signal prior to the CD19CAR? What gene precedes the CAR?

The design of *TRAC* CAR-T followed previous research¹. CAR molecule expression used endogenous promoter of V genes of TCR α which linked *TRAC* locus to construct V-J-C TCR α sequence by splicing in T cells. We added a P2A cleavage site to avoid endogenous TCR α V-J sequence affecting CAR molecule.

7. Why were irradiated Nalm6 needed for measuring MLR? This is unclear.

Nalm6 was required to stimulate the CAR-T cells to proliferate. However, considering its growth might affect the proliferation rate of CAR-T cells, Nalm6 should be irradiated.

8. For Figure 1, as mentioned in one of the bullet points above, comparisons with a non TRAC CAR would be useful. Are MFIs similar, for example? Can the authors comment on the maximum lysis of NALM-6 seen in Fig1D and how this relates to similar CARs in the literature? It would be interesting to see whether the TRAC-inserted CAR has a slightly lower upper limit, making further decreases due to CsA difficult to quantify. Are cells in Fig 1F gated on just the CAR T cells (the ~80% in the population)?

As answered in point 1, the comparison between regular CAR-T (transduced by lentivirus, LV CAR-T) and our AAV CAR-T were performed (Fig. R1, 16).

As the design of our CAR-T followed Eyquem et.al, we would like to compare the killing capability between our CAR and theirs¹ (Fig. R5). As is shown, our CAR showed less cytotoxicity (~2 fold) compared with theirs', probably due to the T cell donor variation. However, the trend is similar, which showed dose-dependent cytotoxicity enhancement.

Fig. R5: Killing assay of CAR-T cells towards tumor cell line Nalm6 in Eyquem (A) and our paper (B).

For the question “Are cells in Fig 1F gated on just the CAR T cells (the ~80% in the population)?”. We are sorry to not clearly clarify in previous version of manuscript. The cells in Fig 1F are the sorted CAR-T cells (red rectangle in figure R6). This was also clarified in our revised paper.

Fig. R6: Representative flow plots of CAR manufacture. Sorted CAR-T cells mean the cell subset in red rectangle.

9. For Figure 2, are the values for WT + vs -CsA and IRU + vs -CsA statistically significant in 2D? The authors only analyzed WT vs IRU and do not show whether CsA has an actual effect within groups.

Thanks for reminding us that the comparison within groups was equally important. The statistical data has been added, not only in Figure 2, but all figures throughout the paper.

10. For Figure 3, do the authors have a Day 0 image of the co culture? Also, it's curious to see that CsA does not eliminate the recipient T cells completely/or as much as the WT CAR T. Can the authors comment on why this would be the case? Finally, it is worth noting too that donor T cell allo reactivity should also exist against recipient T cells. From the co culture experiments, this does not seem to be observed?

Thanks for reminding us to add Day 0 image, to make the figure more completed. We have added them (exemplified in Fig. R7) in all Fig. 3 and extended data Fig. 5.

Fig. R7: Representative flow plots of MLR assay.

For questions “Also, it's curious to see that CsA does not eliminate the recipient T cells completely/or as much as the WT CAR T. Can the authors comment on why this would be the case?”. As our experiments were conducted in interleukin free medium and WT CAR-T cells could not be stimulated with Nalm6 in the presence of CsA to produce IL-2 to support cell survival, all cells in WT CAR-T (+CsA) group cannot survive and proliferate. Over 99% of WT CAR-T and recipient T cells were eliminated in the presence of CsA. However, IRU CAR-T could secrete low level of IL-2 in the presence of CsA (Fig. 1F) to support RTCs survival, which may delay RTCs eliminations in this assay.

Regarding the concern of donor T cell allo reactivity, we previously thought it could be omitted as the TCR of donor T cell has been knocked out. However, this is a very important question of whether our IRU CAR-T would trigger GvHD. Thanks so much to remind us that the CAR-T therapy we raised will be used as a type of universal CAR-T, it is indeed pivotal to evaluate GvHD in this system, although some previous clinical researches have approved it⁷. Therefore, an *in vitro* and *in vivo* experiments were designed (added as Extended data Fig. 3).

In *in vitro* assay, IRU CAR-T cells were cocultured with irradiated recipient PBMCs for 4 days. Then, the release of IFN γ from CAR-T cells that represented the cytotoxicity against recipient PBMCs was detected (Fig. R8A, B; added as Extended data Fig. 3A, B). Comparing CAR-T cells that with functional *TRAC* locus (conventional CAR-T group, made by lentivirus infection), IRU CAR-T cells had significantly lower frequency of IFN γ producing cells. This result indicated the significantly inhibited alloreactive CAR-T cell responses against recipient T cells. Further, the addition of CsA abolished GvHD as the activation of T cells were inhibited. In *in vivo* assay, conventional CAR-T and IRU CAR-T cells were injected in NOD-*Prkdc^{scid} Il2rg^{em1}/Smoc* mice (NSG) mice. The weight and GvHD score (assessed by parameters including posture, activity, fur, skin and weight loss as previously described⁸) of mice were recorded (Fig. R8C, D; added as Extended data Fig. 3C, D). The *in vivo* experiments excluded the circumstances with CsA, as CsA might induce GvHD-like symptoms such as weight change to interfere the outcome interpretation. Compared to IRU CAR-T group, mice injected with conventional CAR-T cells significantly lost weight and scored higher in GvHD index after 5 weeks. The *in vivo* result consistently supported that the GvHD induced by IRU CAR-T cells would significantly decrease due to the lack of TCR expression.

Fig. R8: GvHD assessment of IRU CAR-T cells *in vitro* and *in vivo*. **A**, **B**, Conventional/IRU CAR-T cells were coculture with recipient PBMCs for 4 days.

The secretion of IFN γ was detected by FACS. Representative FACS flow plots (**A**) and bar graph (**B**) of the secretion of IFN γ from conventional/IRU CAR-T cells. **C**, **D**, Conventional/IRU CAR-T cells were injected into NSG mice. The weight (**C**) and GvHD score (**D**) of each mouse was assessed every 1 week.

11. For Figure 4, why is the IRU CAR-T (-CsA) group dropped from the comparisons?

The experiment was designed to exclude the IRU (-CsA) group in previous version, as the novel universal CAR-T was expected to use in the setting with CsA, but not without CsA, in the presence of recipient immune system. However, after careful consideration, we accepted that the IRU (-CsA) control group is necessary to be added in a more scientific perspective.

Fig. R9 (all added in Fig. 4, Extended Data Fig. 6) shows the tumor burden and survival of the IRU (-CsA) group in the mouse model in Fig. 4. Other relative data have been added in the corresponding figures in Fig. 4 and Extended Data Fig. 6. In these updated data, IRU CAR-T (-CsA) group showed a similar trend of WT CAR-T (-CsA) group.

Fig. R9: IRU CAR-T could not retain anti-leukemia functions in the presence of RTCs *in vivo* without CsA. **A.** Tumor burden (average radiance, ph/s) in IRU CAR-T (-CsA) (n=5 mice per group). **B.** Kaplan-Meier analysis of the mice survival in mice treated with WT CAR-T cells (+/-CsA) and IRU CAR-T cells (+/-CsA) (n=5 mice per group).

In addition, to confirm the comparability and consistency of the model with previous data, 3 additional mice in WT (+/- CsA) and IRU (+CsA) groups were performed simultaneously as internal reference (green line in each graph, No. 6, 7, and 8). As shown, this bench generally followed the similar trend in previous version, and IRU (-CsA) group showed similar trend like WT (-CsA) group (Fig. R10).

Fig. R10. Tumor burden (average radiance, ph/s) in WT CAR-T cells (+/-CsA) and IRU CAR-T cells (+/CsA) were performed along with Fig. R5 as internal reference (green line in each graph, No. 6, 7, and 8).

- 1 Eyquem, J. *et al.* Targeting a CAR to the TRAC locus with CRISPR/Cas9 enhances tumour rejection. *Nature* **543**, 113-117, doi:10.1038/nature21405 (2017).
- 2 Feucht, J. *et al.* Calibration of CAR activation potential directs alternative T cell fates and therapeutic potency. *Nat Med* **25**, 82-88, doi:10.1038/s41591-018-0290-5 (2019).
- 3 Turtle, C. J. *et al.* CD19 CAR-T cells of defined CD4+:CD8+ composition in adult B cell ALL patients. *J Clin Invest* **126**, 2123-2138, doi:10.1172/JCI85309 (2016).
- 4 Yang, X. *et al.* Impact of cyclosporine-A concentration in T-cell replete haploidentical allogeneic stem cell transplantation. *Clin Transplant* **32**, e13220, doi:10.1111/ctr.13220 (2018).
- 5 de Kort, E. A. *et al.* Cyclosporine A trough concentrations are associated with acute GvHD after non-myeloablative allogeneic hematopoietic cell transplantation. *PLoS One* **14**, e0213913, doi:10.1371/journal.pone.0213913 (2019).
- 6 Li, X., Daniyan, A. F., Lopez, A. V., Purdon, T. J. & Brentjens, R. J. Cytokine IL-36gamma improves CAR T-cell functionality and induces endogenous antitumor response. *Leukemia* **35**, 506-521, doi:10.1038/s41375-020-0874-1 (2021).
- 7 Qasim, W. *et al.* Molecular remission of infant B-ALL after infusion of universal TALEN gene-edited CAR T cells. *Sci Transl Med* **9**, doi:10.1126/scitranslmed.aaj2013 (2017).
- 8 Cooke, K. R. *et al.* An experimental model of idiopathic pneumonia syndrome after bone marrow transplantation: I. The roles of minor H antigens and endotoxin. *Blood* **88**, 3230-3239 (1996).

Reviewer #3 (expert in T cell signalling):

This paper deals with strategies to avoid GVH and HVG disease when using CAR T cells. The authors used CRISPR/Cas9 to knock the calcineurin A chain (CNA), either WT or a mutant that can't bind to cyclosporin A/cyclophilin A plus a CAR, into the TCR α locus. This accomplishes three things in successfully mutated cells: endogenous TCR expression is lost, CAR expression is gained, and the cells overexpressing the mutated CNA are resistant to the immunosuppressive effects of cyclosporin A. They show that such human primary T cells transferred to tumor-bearing NSG mice can persist and be effective in the presence of cyclosporin A, and can be eliminated when the mice have allogenic T cells and administration of cyclosporin A is discontinued. Overall, the data are consistent and support the potential utility of such an approach. The paper, however, is poorly written, inadequately documented, and the explanations of the results and discussion incomplete or missing.

1. In Fig. 1J, many genes are lower in IRU CAR-T vs WT CAR-T in the absence of CsA. Why? CNA is overexpressed in the transduced cells (the endogenous gene isn't silenced), so is it possible that the IRU mutation is affecting calcineurin functions other than just binding to CyPA? Does it then act as a dominant negative? Some attempt should be made to confirm the RNA-seq profile (i.e. qPCR for message and determination of protein for some genes, like granzyme B). This certainly warrants a discussion.

Thanks for raising a very important question: could mCNA affect T cell effector functions differently compared to WT CNA? (added as Extended data Fig. 2) To address this issue, we firstly expressed WT CAR-T and IRU-CAR-T in Jurkat-NFAT-GFP reporter cell line, then stimulated these cells with Nalm6 cell line without CsA to compare the GFP expression level. As reviewer3 mentioned, the endogenous CNAs were unaffected but WT or mutant CNAs were overexpressed in Jurkat-NFAT-GFP reporter cell line.

We found the frequency of GFP⁺ cells in WT CAR-T-Jurkat-NFAT-GFP cell line were higher than that in IRU-CAR-T-Jurkat-NFAT-GFP cell line in the absence of CsA, when stimulated by Nalm6 (Fig. R1A; added as Extended data Fig. 2A). We hypothesized that this phenomenon is due to the less efficient dephosphorylation of NFAT in mCNA compared to WT CNA. To confirm this, we applied phosphor-flow to detect the level of NFAT phosphorylation in primary T cells. Upon the stimulation of Nalm6 cell line, we also found that the MFI of the phosphorylated NFATc1 in WT CAR-T cells was lower than that in IRU- CAR-T cells without CsA, but the MFI of the phosphorylated NFATc1 in WT CAR-T cells was higher than in IRU- CAR-T cells with CsA (Fig. R1B, C; added as Extended data Fig. 2B, C). Collectively, these data indicated that mCNA had less efficient for NFAT dephosphorylation which affected downstream of NFAT signaling pathway and T cell effector functions.

To confirm this conclusion, we investigated the expression level of downstream genes of NFAT signaling pathway (Fig. R2). We found that the trend of expression

level of downstream genes, such as *Il2*¹, *Il3*¹, *Il5*², *PCNA*^{3,4}, *CSF2*⁵, *Ifng*¹, *Gzmb*¹, and *TNF*¹ followed the similar trend as the level of NFATc1 phosphorylation, i.e., the expression of these genes were lower in IRU CAR-T cells than that in WT CAR-T cells without CsA. Taken together, CAR-T cells overexpression mCNA will affect T effector functions by regulating NFAT dephosphorylation in the absence of CsA.

Fig. R1: WT and mCNA have different deficiencies of dephosphorylation of NFAT signal. **A**, GFP⁺ WT CAR-T-Jurkat-NFAT-GFP cells were significantly higher than GFP⁺ IRU CAR-T-Jurkat-NFAT-GFP cells, upon the stimulation of Nalm6. **B,C**, representative histogram (**B**) and bar graph of MFI (**C**) of the phosphorylated NFATc1 expression in primary WT/IRU CAR-T cells.

For the validation of RNA-seq, we thank for reminding the importance of hands-on experiments to validate bioinformatic data. Following the same protocol, CD8⁺ WT/IRU CAR-T (+/- CsA) cells were obtained, and their cDNA were extracted. qPCR assays were performed (Fig. R2, added in manuscript as Fig. 1K), and the gene expression were calculated using equation $2^{-\Delta t}$ with the Ct of housekeeping gene $\beta 2m$ as standard. As shown, qPCR followed the similar trend of RNA-seq, which further confirmed the accuracy of RNA-seq.

Fig. R2. qPCR validation of representative genes in bulk RNA-seq.

From the qPCR validation assay, we found that the trend of genes expression level, such as the NFAT-regulated activation, survival and proliferation (e.g. *Il2¹*, *Il3¹*, *Il-5²*, *PCNA^{3,4}*, *CSF2⁵*, *CDC25A⁶*, *CDK4^{7,8}*) and NFAT-regulated cytotoxicity (e.g. *Ifng¹*, *Gzmb¹*, and *Tnf¹*) were similar in that of RNA-Seq data. Generally, genes that were positively related to proliferation were highest in WT (-CsA) group, and gradually decreased in IRU (-CsA). The addition of CsA significantly diminished the expression of such genes in WT CAR-T groups, but not in IRU CAR-T groups. In contrast, the apoptosis-related genes (e.g. , *ATM⁹* and *NTRK1¹⁰*) were up-regulated in WT (+CsA) group, rather than any other three groups. These results consistently showed that IRU CAR-T cells could resist the CsA and retain their effector function in the perspective of RNA level.

2. In Supp. Fig.2F, how were the averages calculated? They don't reflect the data. For example, in the first bar for IL-2 the 3 blue dots are all below the average. There are other examples in this figure.

We are badly sorry for these mistakes. We have corrected all of them.

3. In Fig. 3C the decrease in Donor CAR-T cells is seen only at one time point, 9 days. The experiment should show at least one more time point, such as 12 days, to confirm the loss of CAR-T. Also, a control without RTC under the conditions, which differ a bit from 1E, should be shown.

Thanks for helping us to obtain a more completed data, we re-performed the experiment and observed till day 12 (Fig. R3, added as Fig. 3B, C). As shown in Fig. R3B, donor CAR-T percentage and cells, and recipient T cells displayed similar trend in previous version, revealing the robust of the experimental system. Also, a control without RTC under the condition was re-performed (Fig. R3A, added as extended data Fig. 5A), showing a similar trend with Fig. 1E

Fig. R3: MLR assay to validate the survive of IRU CAR-T. (A) CAR-T proliferation without recipient T cells. (B) curve chart showing the percentage of donor CAR-T cells at day 3, 6, 9, and 12 (left) and curve charts of absolute cell counts of donor CAR-T cells (middle) and recipient T cells (right).

4. In Fig. 2D at 14 days (and 21 days to a lesser extent) it appears that the % of IRU CAR-T cells in the bone marrow is actually greater in the presence of CsA. And in Extended Data 2C they are elevated at 10 days (although the error bars are large). Is that because CsA has an affect on the numbers of endogenous murine bone marrow cells? Did their number decrease?

We are really sorry for the mistakes in previous version. Extended Data 2C and Fig. 2D are exactly the same data, just different forms to be shown. The elevated IRU CAR-T cells you mentioned at day 10 was day 14 actually, as the number and axis were a little mismatched. We have corrected it.

However, to address the concern of whether CsA has an effect on the numbers of endogenous murine bone marrow cells, we agreed that the absolute cell numbers were necessary to make the data more evident. Fig. R4 (added as Extended Fig. 7) showed the bone marrow cells for all the mouse model used in this manuscript. No significant was observed to indicate the difference among these mice in different models.

Fig. R4: Bone marrow cells in mouse ALL model, mouse ALL model with RTCs, and mouse ALL model with rechallenged RTCs.

5. Throughout the P values in figure bar graphs are shown comparing WT to IRU CAR-T (see 1F, 2D as examples). What's more important to show, though, is the intra-group comparison of (-CSA) vs. (+CSA). It would also make it easier to compare the behavior of the two groups of CAR-Ts if they were shown in the figure in this order (left to right): WT CAR-T (-CSA), WT CAR-T (+CSA), IRU CAR-T (-CSA), IRU CAR-T (+CSA).

Thanks for reminding us that the comparison within groups was equally important. The statistical data has been added. Also, the order of column has been changed (e.g. Fig. 1F), which definitely made everything easier.

6. The statement on line 185-186 that "the cytotoxic functions of the detectable residual WT CAR-T (+CSA) cells were significantly impaired as quite low frequencies of cytokine-secreting cells were detected..." is misleading. As noted, they measured cytokine production, not CTL activity.

Thanks for helping us to make statement more precisely. It has been corrected as “the capability of cytokine production of the detectable residual WT CAR-T (+CSA) cells were significantly impaired as quite low frequencies of cytokine-secreting cells were detected...”

7. Fig. 3 is inadequately explained and very difficult to follow. Because they want to set up an allogenic MLR it seemed at first that this was being done by mismatching HLA-A2, but that made no sense because it was the donors that were A2-. It seems, rather, than A2 was just being used as a marker so that the donor and recipient cells could be distinguished by flow. That should be made clear. On line 198 it is written "IRU CAR-T (+CSA) group showed significantly higher CAR-T percentage" -- compared to what? I imagine they mean the WT CAR-T (+CSA) group. But the percentages are changing in the other groups too, because the RTCs in the absence of CSA are expanding more than the CAR-T. This is not straightforward because in 3 of the groups both the donor CAR-T cells and the RTC are cyclosporin A-sensitive. This confusing experiment requires more discussion.

Thanks for pointing the confusing interpretation. We have tried our best to describe and provided additional data to make it clear, as below

Due to the central role of allogeneic recipient T cells (RTCs) in induction of HvGR¹¹, we established mixed lymphocyte reaction (MLR) models to mimic RTC rejection¹², using HLA-A2 as a marker to distinguish two donors. HLA-A2⁻ donor CAR-T cells were co-cultured with a 4-fold excess of HLA-A2⁺ RTCs with the stimulation of irradiated Nalm-6-FFluc-GFP- β 2m^{KO} cells that depleted MHC-I to avoid nonspecific allogeneic rejection (Fig. 3A).

CAR-T cells in all groups, except WT CAR-T cells (+CsA), could proliferate without RTCs (R3A, Extended data Fig. 5A). However, the addition of RTCs resulted in the different change of percentage and cell number of CAR-T cells in different groups. Prior to co-culturing with donor CAR-T cells, RTCs were primed as described in methods. Primed RTCs displayed higher percentage of IFN γ producing RTCs (Fig. R5, Extended data Fig. 5B), which contributed to the allo-rejection phenomenon in MLR assay. With the addition of RTCs, IRU CAR-T (+CsA) group showed significantly higher CAR-T percentage than the rest of three groups (Fig. R3B, Fig. 3B, C). Compared to WT CAR-T (+CsA) group, the higher percentage of CAR-T cells in IRU

CAR-T (+CsA) was largely due to the incapability of WT CAR-T proliferation with CsA (R3A, Extended data Fig. 5A). Compared to WT/IRU CAR-T cells in the absence of CsA, the higher percentage of CAR-T cells in IRU CAR-T (+CsA) might result from 1) the allo-rejection of RTCs against CAR-T cells, which resulted in the decreasing cell number of CAR-T cells; 2) the increasing number of RTCs further diluted the percentage of CAR-T cells.

Notably, not all RTCs were real allo-reactive T cells. The percentage of real allo-reactive T cells, i.e. the cells that could produce $IFN\gamma$ after priming (Fig. R5, Extended data Fig. 5B), was not high. Meanwhile, IL-2 produced by the interaction between CAR-T cells and Nalm6 cells/alloreactive T cells could promote the proliferation of non-alloreactive RTCs. Therefore, the decrease of CAR-T cells was not correlated linearly with the increase of RTCs.

Fig. R5: The $IFN\gamma$ secretion cell percentage of RTCs after primed by donor PBMCs for 6 days. Left: RTCs that were not primed by donor PBMCs. Right: RTCs that were primed by donor PBMCs.

8. How was the percent of CAR-T cells that were double producers, such as in Fig. 4G and H, calculated? In G for IRU CAR-T (+CSA) the % of TNF⁺ is about 25% and IFN⁺ maybe 8%, yet the TNF⁺IFN⁺ groups is perhaps 15%. The maximum it could be is 8%, and that's if every IFN⁺ cell also produced TNF.

Sorry for the misleading figure in previous version. In our previous version, the IFN⁺, TNF⁺, IFN⁺TNF⁺ cell population indicated the cells in red, blue, and green rectangle, respectively (Fig. R6). So the percentage of IFN⁺TNF⁺ cell subset is irrelative to that of IFN⁺ and TNF⁺ subsets. To avoid delivering such misleading information for more readers, we have changed the ways to show the percentage of IFN⁺, TNF⁺, IFN⁺TNF⁺ cell population and adapted in Fig. 4F-G, as you suggested.

Fig. R6: schema to illustrate the described IFN⁺, TNF⁺, IFN⁺TNF⁺ cell population in previous version.

9. It's not clear how the cytotoxicity assay in Fig. 1D was performed. According to the Methods, line 402, "sorted" cells were used. Sorted on what? The CAR? CD8+?

Sorry for the ambiguous writing. We are sorry to not clearly clarify in previous version of manuscript. The cells in Fig 1D are the sorted total CAR-T cells (red rectangle in figure R7). This was also clarified in our revised paper.

Fig. R7: Representative flow plots of CAR manufacture. Sorted CAR-T cells mean the cell subset in red rectangle.

10 Fig. 4F and supplemental 4D is incomprehensible. What do the numbers mean (percent?). What do the colors mean? What does the heat scale mean?

Fig. 4F and extended data Fig. 4D were another presenting form of Fig. 4G and extended data Fig. 4E, respectively. The heat scale indicated a range of percentage and the colors mean where the average data was. We previously thought the heatmap demonstration made the comparison more visual friendly. However, after careful consideration and discussion, we thought they were misleading and redundant. So we have deleted it.

11. In Fig. 4B why was analysis of IRU CAR-T (-CsA) not shown? It would seem to be an important control that presumably would show that the IRU CAR-T cells were just as susceptible to WT CAR-T cells to rejection.

The experiment was designed to exclude the IRU (-CsA) group in previous version, as the novel universal CAR-T was expected to use in the setting with CsA, but not without CsA, in the presence of recipient immune system. However, after careful consideration, we accepted that the IRU (-CsA) control group is necessary to be added in a more scientific perspective.

Fig. R8 (all added in Fig. 4, Extended Data Fig. 6) shows the tumor burden and survival of the IRU (-CsA) group in the mouse model in Fig. 4. Other relative data have been added in the corresponding figures in Fig. 4 and Extended Data Fig. 6. In these updated data, IRU CAR-T (-CsA) group showed a similar trend of WT CAR-T (-CsA) group.

Fig. R8: IRU CAR-T could not retain anti-leukemia functions in the presence of RTCs *in vivo* without CsA. **A.** Tumor burden (average radiance, ph/s) in IRU CAR-T (-CsA) (n=5 mice per group). **B.** Kaplan-Meier analysis of the mice survival in mice treated with WT CAR-T cells (+/-CsA) and IRU CAR-T cells (+/-CsA) (n=5 mice per group).

In addition, to confirm the comparability and consistency of the model with previous data, 3 additional mice in WT (+/- CsA) and IRU (+CsA) groups were performed simultaneously as internal reference (green line in each graph, No. 6, 7, and 8). As shown, this bench generally followed the similar trend in previous version, and IRU (-CsA) group showed similar trend like WT (-CsA) group (Fig. R9).

Fig. R9. Tumor burden (average radiance, ph/s) in WT CAR-T cells (+/-CsA) and IRU CAR-T cells (+/CsA) were performed along with Fig. R5 as internal reference (green line in each graph, No. 6, 7, and 8).

Minor point:

- 1. Extended Data 2C. The symbols in the key for the top panel (% of total bone marrow cells) don't match those in the plot.**

We are really sorry for the mistakes in Extended Data 2C. Extended Data 2C and Fig. 2D are exactly the same data, just different forms to be shown. The elevated IRU CAR-T cells you mentioned at day 10 was day 14 actually, as the number and axis were a little mismatched. We have corrected it.

- 1 Klein-Hessling, S. *et al.* NFATc1 controls the cytotoxicity of CD8(+) T cells. *Nat Commun* **8**, 511, doi:10.1038/s41467-017-00612-6 (2017).
- 2 Chong, L. K. *et al.* Proliferation and interleukin 5 production by CD8hi CD57+ T cells. *Eur J Immunol* **38**, 995-1000, doi:10.1002/eji.200737687 (2008).
- 3 Kurki, P., Lotz, M., Ogata, K. & Tan, E. M. Proliferating cell nuclear antigen (PCNA)/cyclin in activated human T lymphocytes. *J Immunol* **138**, 4114-4120 (1987).
- 4 Simonett, S. P. *et al.* Identification of direct transcriptional targets of NFATC2 that promote beta cell proliferation. *J Clin Invest* **131**, doi:10.1172/JCI144833 (2021).
- 5 Shi, Y. *et al.* Granulocyte-macrophage colony-stimulating factor (GM-CSF) and T-cell responses: what we do and don't know. *Cell Res* **16**, 126-133, doi:10.1038/sj.cr.7310017 (2006).
- 6 Kittipatarin, C., Li, W., Durum, S. K. & Khaled, A. R. Cdc25A-driven proliferation regulates CD62L levels and lymphocyte movement in response to interleukin-7. *Exp Hematol* **38**, 1143-1156, doi:10.1016/j.exphem.2010.08.010 (2010).
- 7 Romero-Pozuelo, J., Figlia, G., Kaya, O., Martin-Villalba, A. & Teleanu, A. A. Cdk4 and Cdk6 Couple the Cell-Cycle Machinery to Cell Growth via mTORC1. *Cell Rep* **31**, 107504, doi:10.1016/j.celrep.2020.03.068 (2020).
- 8 Shen, T. & Huang, S. The role of Cdc25A in the regulation of cell proliferation and apoptosis. *Anticancer Agents Med Chem* **12**, 631-639, doi:10.2174/187152012800617678 (2012).
- 9 Shiloh, Y. ATM and related protein kinases: safeguarding genome integrity. *Nat Rev Cancer* **3**, 155-168, doi:10.1038/nrc1011 (2003).
- 10 Dadakhujaev, S. *et al.* Interplay between autophagy and apoptosis in TrkA-induced cell death. *Autophagy* **5**, 103-105, doi:10.4161/auto.5.1.7276 (2009).
- 11 Issa, F., Schiopu, A. & Wood, K. J. Role of T cells in graft rejection and transplantation tolerance. *Expert Rev Clin Immunol* **6**, 155-169, doi:10.1586/eci.09.64 (2010).
- 12 Mo, F. *et al.* Engineered off-the-shelf therapeutic T cells resist host immune rejection. *Nat Biotechnol*, doi:10.1038/s41587-020-0601-5 (2020).

REVIEWERS' COMMENTS

Reviewer #1 (expert in CAR T cells and haematological malignancies):

Authors have addressed most criticisms by performing additional experiments that have much improved the quality of the manuscript.

Reviewer #2 (expert in CAR T cells and adoptive cell therapy):

This reviewer was no longer available for a second review. Reviewer #1 was asked to provide a response instead.

Reviewer #3 (expert in T cell signalling):

To their credit, the authors took all of my comments seriously and addressed those that required additional data with experiments. The results support their original conclusions, so I consider them satisfactorily addressed.

RESPONSE TO REVIEWERS' COMMENTS

Reviewer #1 (expert in CAR T cells and haematological malignancies):

Authors have addressed most criticisms by performing additional experiments that have much improved the quality of the manuscript.

Reviewer #2 (expert in CAR T cells and adoptive cell therapy):

This reviewer was no longer available for a second review. Reviewer #1 was asked to provide a response instead.

Reviewer #3 (expert in T cell signalling):

To their credit, the authors took all of my comments seriously and addressed those that required additional data with experiments. The results support their original conclusions, so I consider them satisfactorily addressed.

We would like to appreciate all reviewers for their evaluations and valuable comments